# FlexEvent: Towards Flexible Event-Frame Object Detection at Varying Operational Frequencies

**Dongyue Lu[1,2], Lingdong Kong[1], Gim Hee Lee[1], Camille Simon Chane[3], Wei Tsang Ooi[1,2]**

[1]National University of Singapore    [2]IPAL, CNRS IRL 2955, Singapore
[2]ETIS UMR 8051, CY Cergy Paris University, ENSEA, CNRS, France

⊕ **Project Page & Code:** `flexevent.github.io`

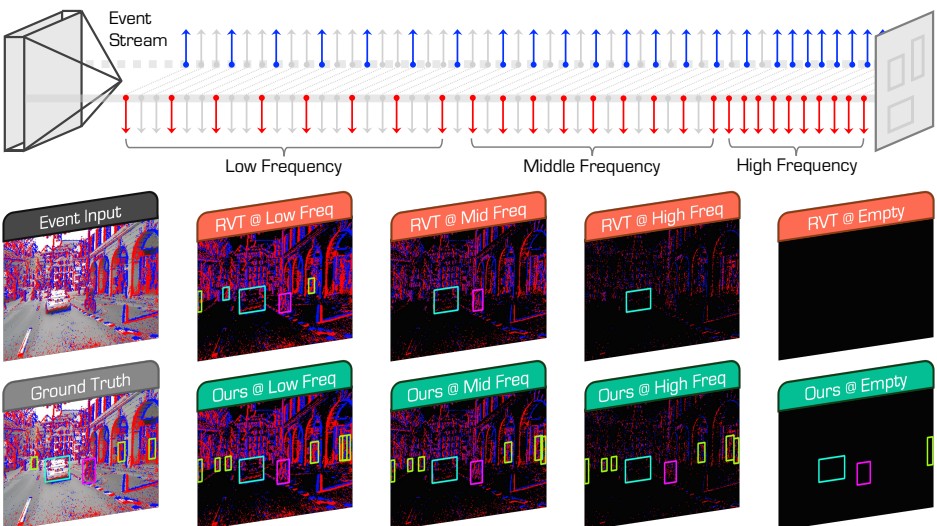

Figure 1: **Illustrative examples of event camera object detection at varying frequencies**. The detection performance of the classic RVT detector [18] tends to drop significantly at higher event operational frequencies. Motivated by this observation, we propose FlexEvent, a robust and flexible event-frame detector that maintains high detection accuracy across a wide range of frequencies (low, middle, high), ensuring strong adaptability in real-world, dynamic sensing environments.

## Abstract

Event cameras offer unparalleled advantages for real-time perception in dynamic environments, thanks to the microsecond-level temporal resolution and asynchronous operation. Existing event detectors, however, are limited by fixed-frequency paradigms and fail to fully exploit the high-temporal resolution and adaptability of event data. To address these limitations, we propose FlexEvent, a novel framework that enables detection at varying frequencies. Our approach consists of two key components: **FlexFuse**, an adaptive event-frame fusion module that integrates high-frequency event data with rich semantic information from RGB frames, and **FlexTune**, a frequency-adaptive fine-tuning mechanism that generates frequency-adjusted labels to enhance model generalization across varying operational frequencies. This combination allows our method to detect objects with high accuracy in both fast-moving and static scenarios, while adapting to dynamic environments. Extensive experiments on large-scale event camera datasets demonstrate that our approach surpasses state-of-the-art methods, achieving significant improvements in both standard and high-frequency settings. Notably, our method

39th Conference on Neural Information Processing Systems (NeurIPS 2025).

maintains robust performance when scaling from 20 Hz to 90 Hz and delivers accurate detection up to 180 Hz, proving its effectiveness in extreme conditions. Our framework sets a new benchmark for event-based object detection and paves the way for more adaptable, real-time vision systems.

# 1 Introduction

Event cameras have garnered significant attention for their ability to capture dynamic scenes with microsecond-level temporal resolution [12]. Unlike RGB cameras, which capture entire frames at fixed intervals, event cameras operate asynchronously, responding to changes in pixel intensity at each location [68]. This low-latency operation reduces motion blur and enables highly energy-efficient sensing, making event cameras ideal for real-time applications [45, 30, 6].

Despite their potential, existing event detectors often fail to fully leverage the high-frequency temporal information [8, 15, 23]. Most approaches align event data with the lower frequency of frames by adopting fixed time intervals between event streams and frame-based annotations [40, 17]. While this simplifies data processing, it inevitably overlooks the rich temporal details embedded in high-frequency event streams. Given that human annotations are often synchronized with slower frame rates, current detectors miss valuable information from higher frequencies, resulting in suboptimal performance when rapid object detection is required in dynamic environments [36, 43].

To address these limitations, we introduce FlexEvent, a novel framework designed to tackle the challenging problem of object detection at varying operational frequencies. Our approach addresses the need for high-frequency detection in fast-changing environments, while adapting to different operational frequencies. We propose two key innovations: **1)** an adaptive event-frame fusion module, and **2)** a frequency-adaptive fine-tuning mechanism.

The first component, **FlexFuse**, addresses the limitations of event data, which often lacks semantic and texture-rich information, especially at higher frequencies [66], by integrating the rich spatial and semantic information from RGB frames with the high-temporal resolution of event streams. It enables high detection accuracy even in fast-moving environments. Furthermore, training on high-frequency event data is computationally expensive and impractical due to the significant human effort required to label such data. FlexFuse mitigates this by sampling event data at varying frequencies, aligning them with the normal frame rate during training, thus maintaining efficiency while preserving the high-frequency benefits at inference time.

The second component, **FlexTune**, enhances the generalization capability of event camera detectors across varying operational frequencies, by generating frequency-adjusted labels for the unlabeled high-frequency data. These labels allow the model to learn from high-frequency event streams without manual annotations, and iterative refinement through self-training ensures that the model remains robust across different motion dynamics and frequency settings. Together, these two components allow for accurate real-time detection in rapid scene changes and adapt to a wide range of operational frequencies, by leveraging the temporal richness of event data and the semantic detail of RGB frames.

Our extensive experiments validate the effectiveness of FlexEvent on multiple large-scale event camera datasets. Our approach consistently outperforms recent detectors across both standard and high-frequency settings. In particular, we achieve mAP **gains of 15.5%, 9.4%**, and **10.3%** over the previous best-performing detectors on *DSEC-Det* [16], *DSEC-Detection* [50], and *DSEC-MOD* [66], respectively. Our model also **maintains 96.2% of its performance** when the operational frequency shifts from **20 Hz** to **90 Hz**, and delivers accurate detection at frequencies **as high as 180 Hz**, proving its robustness under extreme conditions. In summary, our contributions are listed as follows:

• The proposed FlexEvent framework is aimed at tackling the challenging problem of event camera object detection at varying frequencies, being one of the first works to address this challenging problem explicitly in real-world conditions.

• We propose FlexFuse, an adaptive event-frame fusion module that leverages the strengths of both event and frame modalities, enabling more efficient and accurate event-based object detection in dynamic environments.

• We introduce FlexTune, a frequency-adaptive fine-tuning mechanism that generates frequency-adjusted labels and improves generalization across various motion frequencies.

• We demonstrate that our approach achieves promising results across large-scale datasets, particularly in high-frequency scenarios, validating its effectiveness to handle real-world, safety-critical problems.

## 2 Related Work

**Event Camera Object Detection.** Event-based detectors can be broadly split into two approaches: GNNs/SNNs and dense feed-forward models. GNNs build dynamic spatio-temporal graphs by sub-sampling events [15, 47, 36, 43], but they face challenges in propagating information over large spatio-temporal regions, especially for slow-moving objects. SNNs offer efficient sparse information transmission but are often hindered by their non-differentiable nature, complicating optimization processes [9, 8, 59]. Dense, feed-forward models represent the second approach. Initial methods using fixed temporal windows [5, 22, 25, 19] struggle with slow-moving or stationary objects due to their limited capability to capture long-term temporal data. Subsequent work incorporates RNNs and transformers to enhance temporal modeling [40, 69, 34, 18, 39], but these models still lack semantic richness and face difficulties in adapting to variable frequencies and highly dynamic scenarios.

**Event-Frame Multimodal Learning.** Combining events with frame-based data has proven effective in tasks such as deblurring [48, 61], depth estimation [14, 51], and tracking [62, 13]. Early fusion methods perform post-processing combinations [33, 4], but lack feature-level interaction. More recent works propose deeper feature fusion techniques [50, 1, 2], introducing pixel-level attention or temporal transformers for asynchronous processing [66, 32, 16, 3]. Some explore asynchronous multi-modal fusion for flexible inference at different frequencies [32, 14, 60], yet they do not explicitly address high-frequency event data or fully exploit temporal richness. Additionally, balancing contributions from event and frame modalities remains challenging. Unlike these works, FlexEvent introduces a unified fusion framework that adaptively integrates high-frequency event data with semantic-rich frames, achieving robust detection across diverse frequencies while addressing feature imbalance.

**Label-Efficient Learning from Event Data.** Due to limited annotated datasets, label-efficient learning has become an important area for event-based vision. Several studies attempt to reconstruct images from event data [41, 42, 46] or distill knowledge from pre-trained frame-based models [52, 49, 57, 29]. Other approaches use pre-trained models or self-supervised losses [28, 56, 67]. LEOD [55] pioneers object detection with limited labels but does not address high-frequency generalization. A recent state-space model [70] adapts to varying frequencies without retraining but struggles to detect static objects at high frequencies due to its reliance solely on event data. In contrast, FlexEvent is specifically designed to adapt to varying event frequencies, ensuring consistent performance even in scenarios with limited labels, and effectively detecting both stationary and fast-moving objects.

## 3 FlexEvent: A Flexible Event-Frame Object Detector

In this section, we elaborate on the technical details of our FlexEvent framework. We start with the foundational concepts of event data and their representation in Sec. 3.1. We then introduce the **FlexFuse** module in Sec. 3.2, which adaptively fuses event and frame data to enhance detection across varying frequencies. Finally, we detail **FlexTune**, our frequency-adaptive fine-tuning mechanism, in Sec. 3.3, which enables our model to generalize effectively across diverse temporal conditions via adaptive label generation. The overall framework is illustrated in Fig. 2.

### 3.1 Background & Preliminaries

**Event Processing.** Event cameras are bio-inspired vision sensors that capture changes in log intensity per pixel asynchronously, rather than capturing entire frames at fixed intervals. Formally, let $I(x, y, t)$ denote the log intensity at pixel coordinates $(x, y)$ and time $t$. An event $e$ is generated at $(x, y, t)$ whenever the change in log intensity $\Delta I$ exceeds a certain threshold $C$, which is modeled as:

$$\Delta I(x, y, t) = I(x, y, t) - I(x, y, t - \Delta t) \geq C .\tag{1}$$

Each event $e$ is a tuple $(x, y, t, p)$, where $(x, y)$ are the pixel coordinates, $t$ is the timestamp, and $p \in \{-1, 1\}$ denotes the polarity of the event which indicates the direction of the intensity change.

To leverage event data with convolutions, we pre-process events into a 4D tensor $E$ with dimensions [18] representing the polarity, temporal discretization $T$, and spatial dimensions $(H, W)$, respectively.

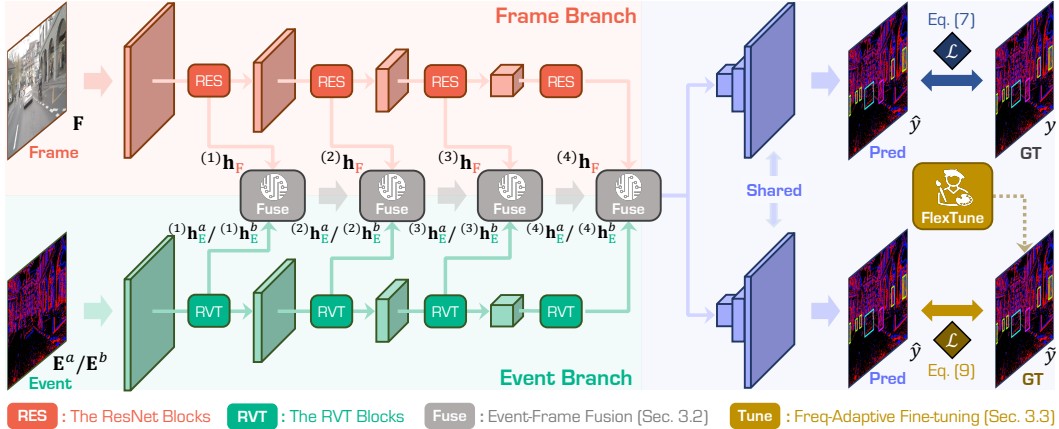

| **RES** : The ResNet Blocks | **RVT** : The RVT Blocks | **Fuse** : Event-Frame Fusion (Sec. 3.2) | **Tune** : Freq-Adaptive Fine-tuning (Sec. 3.3) |

Figure 2: **Framework Overview.** The proposed FlexEvent consists of two branches: Event and Frame. The event branch captures high-temporal resolution data, while the frame branch leverages the rich semantic information from frames (*cf.* Sec. 3.1). These branches are fused dynamically through **FlexFuse**, allowing adaptive integration of event and frame data (*cf.* Sec. 3.2). Additionally, the **FlexTune** learning mechanism ensures robust detection performance across varying operational frequencies (*cf.* Sec. 3.3). Together, these components enable the model to handle diverse motion dynamics and maintain high detection accuracy in varying frequency scenarios.

This representation involves mapping a set of events $\mathcal{E}$ within time interval $[t_1, t_2)$ into the following:

$$E(p, \tau, x, y) = \sum_{e_k \in \mathcal{E}} \delta(p - p_k)\delta(x - x_k, y - y_k)\delta(\tau - \tau_k), \qquad (2)$$

where $\tau_k = \left\lfloor \frac{t_k - t_1}{t_2 - t_1} \cdot T \right\rfloor$. Such a 4D tensor captures event activity in $T$ discrete time slices, yielding a compact representation suitable for 2D operators by flattening the polarity and temporal dimensions.

**Problem Formulation.** Given two consecutive frames $F_1$ and $F_2$ captured at timestamps $t_1$ and $t_2$, our objective is to leverage the event stream over the interval $[t_1, t_2]$ to detect objects at the end timestamp $t_2$. Existing event detectors use fixed time intervals $\Delta t$, limiting adaptability to dynamic environments [40]. Additionally, integrating semantic information from RGB frames remains challenging, affecting performance in complex scenarios [10]. To address this, we propose to synchronize the event data with frames and explore varying training frequencies, leveraging the temporal richness of event cameras to improve detection accuracy.

## 3.2 FlexFuse: Adaptive Event-Frame Fusion

In dynamic environments, object detection systems must adapt to varying motion frequencies [48]. While event cameras excel at capturing rapid changes in pixel intensity, they often lack rich spatial and semantic information provided by frames. To address this limitation and fully leverage the complementary strengths of both modalities, we introduce **FlexFuse**, an adaptive fusion module designed to dynamically combine event data at different frequencies with frames.

**Dynamic Event Aggregation.** Given a dataset $\mathcal{D}$, consisting of sequences of calibrated event data and frame data with a resolution of $H \times W$, along with corresponding bounding box annotations $y$ collected at frequency $a$, we begin by selecting a batch of frame data $\mathbf{F}$ paired with event data $\mathbf{E}^a$, both captured at frequency $a$. To aggregate event data from a higher frequency $b$ (where $b > a$), we divide the time interval $\Delta t^a$ corresponding to $\mathbf{E}^a$ into $b/a$ smaller sub-intervals. From each sub-interval, we obtain a high-frequency event set $\{\mathbf{E}_i^b\}_{i=0}^{b/a}$, as defined in Eq. 2. From this set, we randomly sample one data point[1] $\mathbf{E}^b$. As detections are predicted for the end of $\Delta t^a$, this strategy pairs each frame with the preceding high-frequency events while introducing only millisecond-scale jitter that acts as implicit temporal augmentation, strengthening robustness to real-world synchronization

---

[1]For simplicity, we use $\mathbf{E}^b$ to represent a sample from the set of high-frequency event data $\{\mathbf{E}_i^b\}_{i=0}^{b/a}$, rather than explicitly referencing each individual sample from the event set. The same applies to other frequencies.

noise. Consequently, $(\mathbf{F}, \mathbf{E}^a, \mathbf{E}^b)$ are effectively paired across modalities and frequencies, and this synchronization of image streams with event streams at different frequencies ensures consistent and reliable processing for all subsequent stages.

**Feature Extraction.** Let $\phi_{\mathrm{E}}(\cdot)$ and $\phi_{\mathrm{F}}(\cdot)$ represent the event- and frame-based networks, respectively, where the former employs the RVT [18] for extracting features from event data, and the latter uses ResNet-50 [20] for feature extraction from frames. Both networks are structured into four stages, as shown in Fig. 2. At each scale $i$, we extract features $^{(i)}\mathbf{h}_{\mathrm{E}}^a, {}^{(i)}\mathbf{h}_{\mathrm{E}}^b$ from the event data and $^{(i)}\mathbf{h}_{\mathrm{F}}$ from the frame data. This process is formulated as:

$$^{(i)}\mathbf{h}_{\mathrm{E}}^a =^{(i)} \phi_{\mathrm{E}}(\mathbf{E}^a), {}^{(i)}\mathbf{h}_{\mathrm{E}}^b =^{(i)} \phi_{\mathrm{E}}(\mathbf{E}^b), {}^{(i)}\mathbf{h}_{\mathrm{F}} =^{(i)} \phi_{\mathrm{F}}(\mathbf{F}), \qquad (3)$$

where $^{(i)}\mathbf{h}_{\mathrm{E}}^a$ and $^{(i)}\mathbf{h}_{\mathrm{E}}^b \in \mathbb{R}^{B \times C_{\mathrm{E}} \times H_i \times W_i}$, $^{(i)}\mathbf{h}_{\mathrm{F}} \in \mathbb{R}^{B \times C_{\mathrm{F}} \times H_i \times W_i}$. Here, $i$ denotes the scale, $B$ is the batch size, and $C_{\mathrm{E}}$ and $C_{\mathrm{F}}$ are the dimensions of the event and frame feature maps, respectively.

**Event-Frame Adaptive Fusion.** To effectively fuse the event and frame data, we employ an adaptive fusion that is consistent across different event data frequencies. At each scale $i$, taking the low-frequency event features $^{(i)}\mathbf{h}_{\mathrm{E}}^a$ as an example, we concatenate the feature maps from both the event and frame branches as follows:

$$^{(i)}\mathbf{h}_{\mathrm{shared}}^a = \left[ {}^{(i)}\mathbf{h}_{\mathrm{E}}^a, \ {}^{(i)}\mathbf{h}_{\mathrm{F}} \right] \in \mathbb{R}^{B \times (C_{\mathrm{E}}+C_{\mathrm{F}}) \times H_i \times W_i} . \qquad (4)$$

Inspired by previous work [66, 64], our goal is to dynamically fuse these two modalities in a flexible manner. The proposed FlexFuse module computes adaptive soft weights that regulate the contribution of each branch (event and frame) based on the current input conditions. As shown in Fig. 3, these adaptive soft weights are computed using a gating function, which incorporates learned noise to introduce perturbations for improved adaptability:

$$[^{(i)}\boldsymbol{\alpha}, \ {}^{(i)}\boldsymbol{\beta}] = \mathrm{Softmax}((^{(i)}\mathbf{h}_{\mathrm{shared}}^a \cdot^{(i)} \mathbf{W}) +^{(i)} \sigma \cdot \epsilon), \quad (5)$$

where $^{(i)}\mathbf{W} \in \mathbb{R}^{(C_{\mathrm{E}}+C_{\mathrm{F}}) \times 2}$ is a trainable weight matrix, $^{(i)}\boldsymbol{\alpha}$ and $^{(i)}\boldsymbol{\beta}$ are the adaptive soft weights for the event and frame branches, respectively. Here, $^{(i)}\sigma$ is a learned standard deviation that controls the magnitude of the noise perturbation, and $\epsilon \sim \mathcal{N}(0,1)$ represents a Gaussian noise term. The fused feature map at each scale $i$ is then obtained by applying adaptive soft weights:

$$^{(i)}\mathbf{h}_{\mathrm{fuse}}^a =^{(i)} \boldsymbol{\alpha} \odot^{(i)} \mathbf{h}_{\mathrm{E}}^a +^{(i)} \boldsymbol{\beta} \odot^{(i)} \mathbf{h}_{\mathrm{F}}, \qquad (6)$$

where $\odot$ denotes element-wise multiplication. This fusion process dynamically balances the contribution of each modality based on the input data, allowing for more adaptive feature representation across varying conditions.

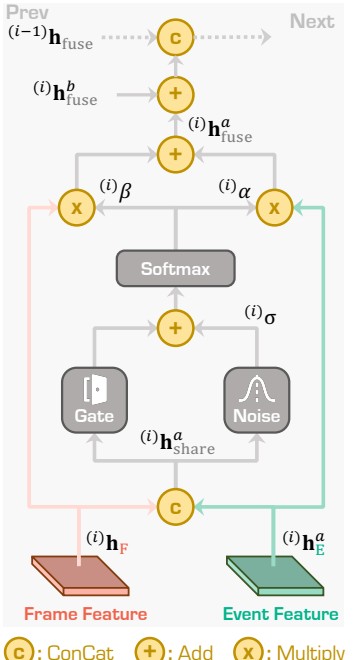

Figure 3: Illustration of the **Flex-Fuse** module. We show a general example of event and frame under frequency $a$ at the $i$-stage.

Then, at each scale $i$, the final feature map combining event data at different frequencies and the frame data is obtained by adding the fused features from different frequencies. Specifically, we combine the fused feature maps as $^{(i)}\mathbf{h}_{\mathrm{fuse}} =^{(i)} \mathbf{h}_{\mathrm{fuse}}^a +^{(i)} \mathbf{h}_{\mathrm{fuse}}^b$. After obtaining the fused feature maps across all scales, the multi-scale features are concatenated and fed into the detection head to produce the predicted bounding box $\hat{\mathbf{y}}$.

**Optimization & Regularization.** In addition to the standard detection loss $\mathcal{L}_{\mathrm{det}}(\mathbf{y}, \hat{\mathbf{y}})$, we introduce a regularization term to ensure balanced utilization of both the event and frame branches. This term penalizes large variations in the soft weights, encouraging a more uniform contribution from both modalities and preventing overfitting to a single branch:

$$\mathcal{L}_{\mathrm{fuse}} = \mathcal{L}_{\mathrm{det}}(\mathbf{y}, \hat{\mathbf{y}}) + \lambda \left( \frac{\mathrm{Var}(\boldsymbol{\alpha})}{(\mathbb{E}[\boldsymbol{\alpha}])^2} + \frac{\mathrm{Var}(\boldsymbol{\beta})}{(\mathbb{E}[\boldsymbol{\beta}])^2} \right), \qquad (7)$$

where $\lambda$ is a weighting factor.

### 3.3 FlexTune: Frequency-Adaptive Fine-Tuning

FlexFuse aggregates information from different frequencies using labeled low-frequency data. To tune the model adaptively to handle diverse frequencies by leveraging both labeled low-frequency data and unlabeled high-frequency data, we design a **FlexTune** learning mechanism, incorporating multi-frequency information into the training process through iterative self-training. This enhances the ability to generalize across varying frequencies, making it more robust in different scenarios.

As shown in Fig. 4, FlexTune consists of two main stages: Low-Frequency Sparse Training to learn foundational knowledge from labeled data, and Cross-Frequency Propagation to transfer and refine the learned knowledge with high-frequency unlabeled data. This iterative mechanism bridges sparse supervision with dense temporal patterns, enabling robust detection under varying frequencies.

**Low-Frequency Sparse Training.** Rather than training solely at frequency $a$, we enhance the model's capability by training it at a higher frequency $b$. To efficiently leverage the available sparse labels, we select only the last event from the high-frequency event set $\{\mathbf{E}_i^b\}_{i=0}^{b/a}$, which corresponds to the labeled timestamp. This allows the model to capture valuable high-frequency temporal information while still utilizing low-frequency labels, improving its temporal understanding and robustness. The training objective is to minimize the detection loss over the sparse labeled data as:

$$\mathcal{L}_{\mathrm{GT}} = \sum \mathcal{L}_{\mathrm{det}}(\mathbf{y}, \hat{\mathbf{y}}), \qquad (8)$$

where the summation is taken over samples $(\mathbf{F}, \mathbf{E}_{b/a}^b, \mathbf{y}) \in \mathcal{D}$.

Figure 4: Illustration of the **FlexTune learning mechanism**. We first train on high-frequency events with sparse low-frequency labels, then we generate and refine high-frequency labels for cyclic self-training across frequencies.

**Cross-Frequency Propagation.** Building on the low-frequency training, we transfer and refine model knowledge to unlabeled high-frequency data through three sequential steps: *High-Frequency Bootstrapping*, *Temporal Consistency Calibration*, and *Cyclic Self-Training*. Together, these steps generate accurate high-frequency pseudo-labels, enabling robust detection in diverse temporal settings.

*High-Frequency Bootstrapping.* For the unlabeled data in $\mathcal{D}$ captured at frequency $b$, the pre-trained model generates high-frequency labels $\tilde{\mathbf{y}}$ by performing inference on the entire high-frequency event set $\{\mathbf{E}_i^b\}_{i=0}^{b/a}$. These generated labels $\tilde{\mathbf{y}}$ serve as pseudo-labels for bootstrapping further training at higher frequencies, improving the model's ability to generalize across different temporal conditions.

*Temporal Consistency Calibration.* To refine high-frequency labels, we enforce temporal coherence through three steps. **(1) Bidirectional Event Augmentation.** Process event streams forward and backward to capture diverse object motions, enhancing recall. **(2) Confidence-Aware Filtering.** Apply Non-Maximum Suppression (NMS) and a low confidence threshold $\tau$ to eliminate duplicates and retain high-potential detections. **(3) Tracklet Pruning.** Link detections across frames using IoU-based tracking $(\tau^{\mathrm{IoU}})$ and prune short tracks $(< \mathbf{L}^{\mathrm{track}})$ to suppress transient noise. This approach ensures that the refined high-frequency labels $\tilde{\mathbf{y}}$ are accurate, temporally consistent, and reliable, improving detection quality in high-frequency data even in the absence of ground truth labels.

*Cyclic Self-Training.* The model is iteratively trained using the refined high-frequency labels $\tilde{\mathbf{y}}$ and the ground truth low-frequency label. The total loss function combines the base training loss and the pseudo-label loss as:

$$\mathcal{L}_{\mathrm{tune}} = \mathcal{L}_{\mathrm{GT}} + \beta \sum \mathcal{L}_{\mathrm{det}}\left(\tilde{\mathbf{y}}, \hat{\mathbf{y}}\right), \qquad (9)$$

where the summation here is taken over high-frequency samples $(\mathbf{F}, \{\mathbf{E}_i^b\}_{i=0}^{b/a-1}, \tilde{\mathbf{y}}) \in \mathcal{D}$, and the coefficient $\beta$ balances the contribution of the high-frequency label loss. The complete FlexEvent framework combines FlexFuse and FlexTune, allowing the model to dynamically fuse event and frame data while adapting to varying frequencies.

Table 1: Comparative study of **state-of-the-art event camera detectors** on the validation set of the *DSEC-Det* [16] dataset. Both event-only and event-frame fusion methods are compared. The **best** and 2nd best scores from each metric are highlighted in **bold** and underlined, respectively.

| Modality | Method | Ref | Venue | mAP | $AP_{50}$ | $AP_{75}$ | $AP_S$ | $AP_M$ | $AP_L$ |
|---|---|---|---|---|---|---|---|---|---|
| E | RVT | [18] | CVPR'23 | 38.4 | 58.7 | 41.3 | 29.5 | 50.3 | 81.7 |
|  | SAST | [39] | CVPR'24 | 38.1 | 60.1 | 40.0 | 29.8 | 48.9 | 79.7 |
|  | SSM | [70] | CVPR'24 | 38.0 | 55.2 | 40.6 | 28.8 | 52.2 | 77.8 |
|  | LEOD | [55] | CVPR'24 | 41.1 | 65.2 | 43.6 | 35.1 | 47.3 | 73.3 |
| E + F | HDI-Former | [31] | arXiv'24 | 46.7 | 69.1 | - | - | - | - |
|  | DAGr-18 | [16] | Nature'24 | 37.6 | - | - | - | - | - |
|  | DAGr-34 | [16] | Nature'24 | 39.0 | - | - | - | - | - |
|  | DAGr-50 | [16] | Nature'24 | 41.9 | 66.0 | 44.3 | 36.3 | 56.2 | 77.8 |
|  | FlexEvent | - | **Ours** | **57.4** | **78.2** | **66.6** | **51.7** | **64.9** | **83.7** |

Table 2: Comparative study of **state-of-the-art event camera detectors** on the test set of *DSEC-Detection* [50]. Both event-only and event-frame fusion methods are compared. The reported results are the **mAP** scores of the [1]Car, [2]Pedestrian, and [3]Large-Vehicle (L-Veh.) classes. The **best** and 2nd best scores from each metric are highlighted in **bold** and underlined, respectively.

| Modality | Method | Venue | Ref | Type | Car | Pedestrian | L-Veh. | Average |
|---|---|---|---|---|---|---|---|---|
| E | CAFR | ECCV'24 | [3] | Event | - | - | - | 12.0 |
| E + F | SENet | CVPR'18 | [21] | Attention | 38.4 | 14.9 | 26.0 | 26.2 |
|  | CBAM | ECCV'18 | [54] |  | 37.7 | 13.5 | 27.0 | 26.1 |
|  | ECA-Net | CVPR'20 | [53] |  | 36.7 | 12.8 | 27.5 | 25.7 |
|  | SAGate | ECCV'20 | [7] | RGB + Depth | 32.5 | 10.4 | 16.0 | 19.6 |
|  | DCF | CVPR'21 | [24] |  | 36.3 | 12.7 | 28.0 | 25.7 |
|  | SPNet | ICCV'21 | [65] |  | 39.2 | 17.8 | 26.2 | 27.7 |
|  | RAMNet | RA-L'21 | [14] | RGB + Event | 24.4 | 10.8 | 17.6 | 17.6 |
|  | FAGC | Sensors'21 | [2] |  | 39.8 | 14.4 | 33.6 | 29.3 |
|  | FPN-Fusion | ICRA'22 | [50] |  | 37.5 | 10.9 | 24.9 | 24.4 |
|  | EFNet | ECCV'22 | [48] |  | 41.1 | 15.8 | 32.6 | 30.0 |
|  | DRFuser | EAAI'23 | [37] |  | 38.6 | 15.1 | 30.6 | 28.1 |
|  | CMX | TITS'23 | [58] |  | 41.6 | 16.4 | 29.4 | 29.1 |
|  | RENet | ICRA'23 | [66] |  | 40.5 | 17.2 | 30.6 | 29.4 |
|  | CAFR | ECCV'24 | [3] |  | 49.9 | 25.8 | 38.2 | 38.0 |
|  | FlexEvent | **Ours** | - |  | **59.3** | **37.4** | **45.5** | **47.4** |

# 4 Experiments

## 4.1 Experimental Settings

**Datasets.** We conduct experiments based on three large-scale datasets: [1]*DSEC-Det* [16], [2]*DSEC-Detection* [50], and [3]*DSEC-MOD* [66]. These datasets comprise 78,344 frames across 60 sequences, 52,727 frames over 41 sequences, and 13,314 frames within 16 sequences, respectively. making them suitable for evaluating event-based object detection methods. We prioritize DSEC-Det [16] as the primary benchmark for comparisons, as it is the largest, most recent, and most comprehensive event-frame perception dataset. We report results on Car and Pedestrian classes to ensure fair comparison with the state-of-the-art method DAGr [16]. For more details, please refer to Appendix Sec. A.1.

**Baselines.** To evaluate our method, we compare it against both state-of-the-art *event-only* and *event-frame fusion* methods. For *event-only* detectors, we compare RVT [18], SAST [39], LEOD [55], and SSM [70], retraining them on DSEC-Det [16] following their respective protocols . For *event-frame fusion* methods, we compare DAGr [16] and HDI-Former [31] on DSEC-Det [16] using results from the original paper and retrain our method on DSEC-Detection [50] and DSEC-MOD [66] using standard settings to compare with CAFR [3] and RENet [66]. For other methods on DSEC-Detection and DSEC-MOD, we reference results reported in the CAFR and RENet papers. For more details, please refer to Appendix Sec. A.2.

**Evaluation Metrics.** We evaluate object detectors using the mean Average Precision (mAP) as the primary metric, along with $AP_{50}$, $AP_{75}$, $AP_S$, $AP_M$, and $AP_L$ from the COCO evaluation protocol [35]. These metrics provide a comprehensive assessment of detection performance across different IoU thresholds and object sizes. Please refer to the Appendix Sec. A.3 for more details.

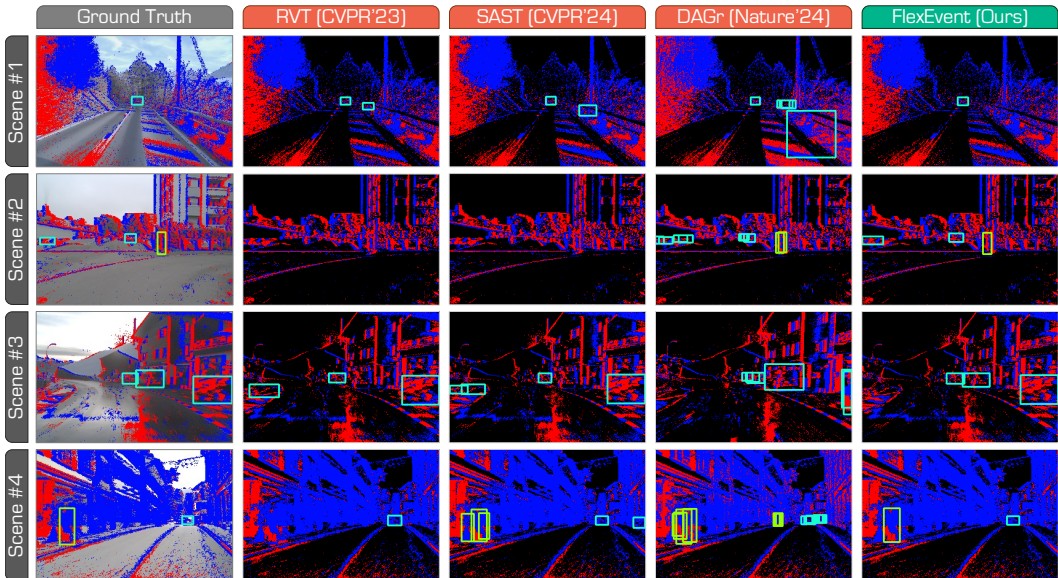

Figure 5: **Qualitative comparisons** of state-of-the-art event camera detectors. We compare FlexEvent with RVT [18], SAST [39], and DAGr [16] on the test set of *DSEC-Det*. Best viewed in colors.

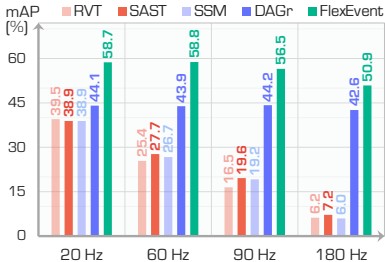

Figure 6: Results of detectors under **varied frequencies** on *DSEC-Det*.

Table 3: Comparisons of **state-of-the-art event-frame fusion detectors** on the test set of the *DSEC-MOD* [66] dataset.

| Modal | Method | Venue | Type | F.mAP | V.mAP | Avg |
|---|---|---|---|---|---|---|
| | SENet [21] | CVPR'18 | | 29.3 | 17.5 | 23.4 |
| | CBAM [54] | ECCV'18 | Attention | 36.2 | 16.4 | 26.3 |
| | ECA-Net [53] | CVPR'20 | | 34.5 | 18.8 | 26.7 |
| | SAGate [7] | ECCV'20 | | 33.6 | 17.8 | 25.7 |
| E+F | DCF [24] | CVPR'21 | RGB + Depth | 32.2 | 18.8 | 25.5 |
| | SPNet [65] | ICCV'21 | | 32.7 | 14.7 | 23.7 |
| | FPN-Fusion [50] | ICRA'22 | | 32.3 | 15.0 | 23.7 |
| | EFNet [48] | ECCV'22 | RGB + Event | 35.3 | 18.1 | 26.7 |
| | RENet [66] | ICRA'23 | | 38.4 | 19.6 | 29.0 |
| | FlexEvent | **Ours** | | **48.6** | **25.2** | **36.9** |

**Implementation Details.** Our training follows YOLO-X [63], using the standard detection loss consisting of IoU loss, classification loss, and regression loss, plus a lightweight regularizer that balances the contributions of each modality. The model is trained for 100,000 iterations with a batch size of 8 and a sequence length of 11, using a learning rate of 1e-4. Experiments are conducted on two NVIDIA RTX A5000 GPUs with 24GB memory, with the entire training process completed in approximately one day. Due to space limits, more details are in Appendix Sec. A.4.

### 4.2 Comparisons to State of the Arts

**Comparisons with Event-Only Detectors.** We compare FlexEvent with state-of-the-art event-only detectors, including RVT [18], SSM [70], SAST [39], and LEOD [55], as shown in Tab. 1. Our model substantially outperforms all these methods, with the gap widening at higher event frequencies. Event-only methods struggle to maintain detection accuracy at higher frequencies because they lack rich semantic cues. In contrast, we overcome these limitations through the FlexFuse module, which integrates RGB data to compensate for the lack of semantic richness in the event stream. By fusing both event and frame data, we excel in complex and dynamic environments, achieving superior detection accuracy where event-only methods fall short.

**Comparisons with Multi-Modal Detectors.** We compare FlexEvent with multimodal event-camera object detection methods such as DAGr [16], HDI-Former [31], CAFR [3], and RENet [66], which fuse event data with other sensor inputs to improve detection accuracy. The results are shown in Tab. 1, Tab. 2 and Tab. 3. While these methods enhance performance over event-only approaches,

Table 4: **Comparative efficiency analysis** of event detectors on *DSEC-Det* [16]. We report the **inference times** of event detectors at various frequencies, measured in **milliseconds (ms)**.

| Method | Size (M) | Operational Frequency | | | |
|---|---|---|---|---|---|
| | | 20.0 Hz | 36.0 Hz | 90.0 Hz | 180 Hz |
| RVT [18] | 18.5 | 9.20 ms | 7.93 ms | 7.19 ms | 6.77 ms |
| SAST [39] | 18.9 | 14.06 ms | 12.37 ms | 11.52 ms | 11.10 ms |
| SSM [70] | 18.2 | 8.79 ms | 7.71 ms | 6.90 ms | 6.54 ms |
| DAGr-50 [16] | 34.6 | 73.35 ms | 55.11 ms | 45.29 ms | 43.89 ms |
| FlexEvent | 45.4 | 14.27 ms | 13.00 ms | 12.47 ms | 12.37 ms |

Table 5: **Ablation on** FlexFuse and FlexTune learning mechanism on *DSEC-Det* [16]. Symbol ♦ denotes the use of interpolated ground truth labels at high frequencies in FlexTune.

| Tune | Fuse | Frequency (Hz) | | | | | | Avg |
|---|---|---|---|---|---|---|---|---|
| | | 20.0 | 36.0 | 45.0 | 60.0 | 90.0 | 180 | |
| ✗ | ✗ | 53.2 | 52.0 | 49.4 | 45.9 | 38.8 | 22.9 | 43.7 |
| ✓ | ✗ | 54.6 | 54.3 | 53.3 | 50.7 | 44.6 | 30.4 | 48.0 |
| ♦ | ✓ | 54.9 | 57.8 | 57.2 | 56.1 | 53.7 | 48.3 | 54.7 |
| ✗ | ✓ | **58.0** | 59.6 | 59.0 | 57.6 | 54.8 | 49.2 | 56.4 |
| ✓ | ✓ | 57.4 | **60.1** | **59.5** | **58.8** | **56.5** | **50.9** | **57.2** |

they struggle to adapt to varying operational frequencies and often exhibit inadequate feature fusion in dynamic environments. Our approach addresses these limitations by dynamically balancing the contributions of event and frame data. This flexible combination, along with the ability to generalize across different temporal resolutions, enables our method to excel in high-frequency event-based detection scenarios, surpassing state-of-the-art approaches.

**Generalization on High-Frequency Data.** A key contribution of FlexEvent is its generalization capability across various operational frequencies, particularly in high-frequency scenarios. We evaluate this by testing detection performance at different temporal offsets, $\frac{i}{n}\Delta T$, where $n = 10$, $i = 0, ..., 10$, and $\Delta T = 50$ms. Ground truth labels are generated by linearly interpolating object positions between frames following DAGr [16]. In this setting, event-based methods are tested across multiple time durations, while event-frame fusion methods process one RGB frame followed by event data of varying time durations. As shown in Fig. 6, existing methods degrade significantly at higher frequencies due to fixed temporal intervals and limited ability to capture fast scene changes, while our method achieves consistent, strong performance.

**Comparisons on Efficiency.** In Tab. 4, we compare inference times and parameter counts using an NVIDIA A5000 24GB GPU and an AMD EPYC 32-Core Processor. Although FlexEvent comprises slightly more parameters overall, its speed matches SAST [39] and clearly surpasses DAGr [16] at all tested frequencies. As FlexTune is performed *off-line*, the high-frequency generalizability is already propagated in the model and introduces no runtime overhead. Our lightweight fusion head contributes only $\sim 1.5$M parameters, while the event–frame backbone adds a further $\sim 44$M that is already highly optimized and incurs virtually no extra latency. This confirms our efficiency and real-time suitability.

**Qualitative Assessments.** We provide visual comparisons of FlexEvent and other state-of-the-art methods under different event operation frequencies, as shown in Fig. 5. Unlike RVT [18] and DAGr [16], which often miss or misinterpret critical objects, our model consistently detects objects with high accuracy, even in challenging cases involving fast-moving vehicles and occluded pedestrians. This robustness is also evident in examples of Fig. 7, where our method detects a pedestrian missed by RVT [18] due to sparse event data at high frequencies. For more detailed analyses, visualizations, and failure cases, please refer to Appendix Sec. C.

## 4.3 Ablation Study

**Analysis of FlexFuse.** We assess the effect of integrating frame data via FlexFuse. As shown in Tab. 5, adding frame information boosts average mAP from $43.7\%$ to $56.4\%$, with greater gains at higher frequencies where event data alone becomes sparse. Our adaptive gating fuses event and frame features more effectively than simple *Add*, *Concat*, or vanilla *Attention* (Fig. 8), achieving the best accuracy across all frequencies and providing robust representations under diverse conditions.

**Analysis of FlexTune.** In parallel, we assess FlexTune, focusing on scenarios where event data are sparse and fast-evolving. Without it, performance at high frequencies collapses, as seen in the first row of Tab. 5 where mAP at $180$ Hz is only $22.9\%$. However, activating FlexTune raises this score to $30.4\%$, demonstrating the value of iterative self-training and refinement for frequency adaptation. We also test training with interpolated high-frequency labels, but this approach struggles with objects that appear or disappear rapidly, reducing recall of the label. By contrast, FlexTune updates the model with refined, temporally consistent labels, improving detection quality.

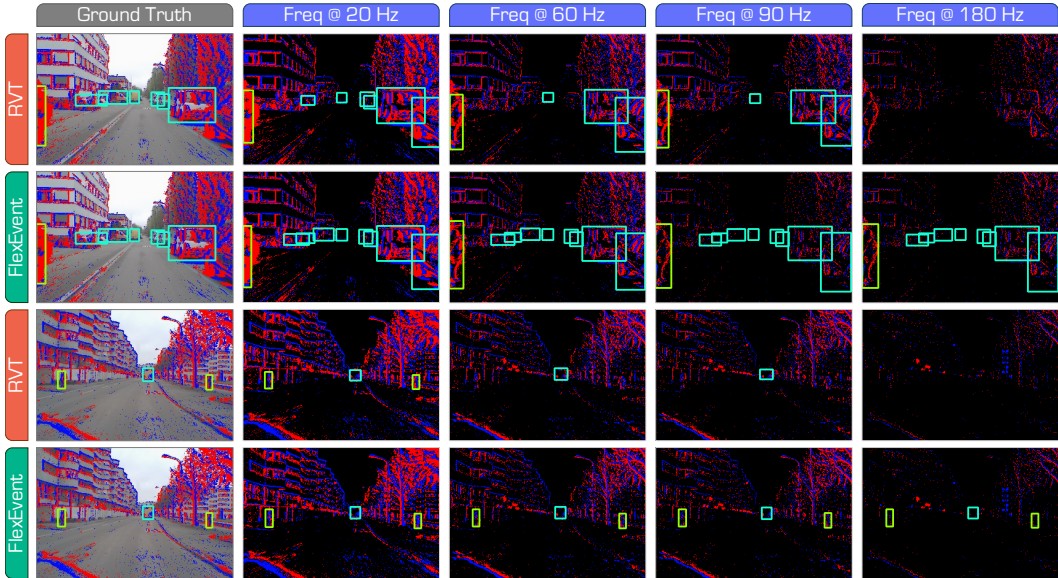

Figure 7: **Qualitative assessment** of FlexEvent and RVT [18] under different operation frequencies.

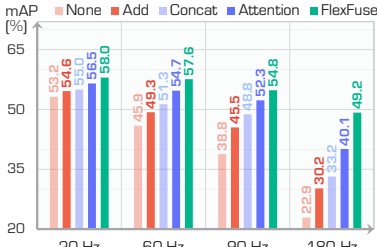

Figure 8: Ablation on event-frame **fusion strategies** under different frequencies on *DSEC-Det* [16].

Table 6: **Ablation study on hyperparameters.** $\tau^c$, $\tau^p$ denotes the confidence threshold for *Car* and *Pedestrian*, respectively. $\tau^{iou}$ denotes the IoU threshold when filter by tracking, $\mathbf{L}^{track}$ denotes the minimum track length. The reported results are the **mAP** scores on the test set of *DSEC-Det* [16].

| $\tau^c$ | $\tau^p$ | $\mathbf{L}^{track}$ | $\tau^{iou}$ | \multicolumn{6}{c}{Frequency (Hz)} | Avg |
| | | | | 20.0 | 36.0 | 45.0 | 60.0 | 90.0 | 180 | |
|---|---|---|---|---|---|---|---|---|---|---|
| 0.6 | 0.3 | 10 | 0.8 | 56.5 | 57.2 | 57.1 | 56.7 | 54.5 | 49.2 | 55.2 |
| 0.6 | 0.3 | 10 | 0.6 | 56.7 | 57.9 | 57.7 | 57.0 | 54.3 | 47.0 | 55.1 |
| 0.6 | 0.3 | 8 | 0.6 | 56.3 | 59.1 | 58.8 | 58.4 | 56.2 | **51.2** | 56.7 |
| 0.6 | 0.3 | 6 | 0.6 | 57.3 | 59.9 | 59.3 | 58.5 | 55.7 | 48.8 | 56.6 |
| 0.6 | 0.6 | 6 | 0.6 | **57.4** | **60.1** | **59.5** | **58.8** | **56.5** | 50.9 | **57.2** |
| 0.8 | 0.8 | 6 | 0.6 | 56.6 | 58.9 | 58.4 | 57.4 | 55.6 | 50.2 | 56.2 |

**Hyperparameter Tuning.** We investigate key hyperparameters in the FlexTune mechanism, specifically, the confidence thresholds for cars $\tau^c$ and pedestrians $\tau^p$, the IoU threshold for bounding-box association $\tau^{iou}$, and the minimum track length for temporal refinement $\mathbf{L}^{track}$. As shown in Tab. 6, lowering confidence thresholds can improve recall by admitting more detections, but it risks increasing false positives and thus reducing overall precision. Conversely, setting overly strict thresholds, such as a higher IoU requirement or confidence level, might filter out potential detections, lowering recall. An intermediate setting of $\tau = 0.6$ for both classes, with a track length of 6 and an IoU threshold of 0.6, emerges as the best trade-off, balancing recall and precision across both low- and high-frequency conditions. These moderate, carefully tuned parameters ensure that FlexEvent remains robust and accurate in a range of real-world scenarios.

## 5 Conclusion

This paper introduces FlexEvent, an event-camera detection framework designed to operate across varying frequencies. By combining FlexFuse for adaptive event-frame fusion and FlexTune for frequency-adaptive learning, we leverage the rich temporal information of event data and the semantic detail of RGB frames to overcome the limitations of existing methods, providing a flexible solution for dynamic environments. Extensive experiments on large-scale datasets show that our approach significantly outperforms state-of-the-art methods, particularly in high-frequency scenarios, demonstrating its robustness and adaptability for real-world applications.

## Acknowledgments

The authors would like to sincerely thank the Program Chairs, Area Chairs, and Reviewers for the time and effort devoted during the review process.

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

# Appendix

## A  Additional Implementation Details

In this section, to facilitate future reproductions, we elaborate on the necessary details in terms of the datasets, evaluation metrics, and implementation details adopted in our experiments.

### A.1  Datasets

In this work, we develop and validate our proposed approach on the large-scale DSEC dataset [17]. DSEC serves as a high-resolution, large-scale multimodal dataset designed to capture real-world driving scenarios under various conditions. It combines data from stereo Prophesee Gen3 event cameras with a resolution of $640 \times 480$ pixels and FLIR Blackfly S RGB cameras operating at 20 FPS, enabling high-fidelity capture of dynamic scenes. To align the RGB frames with the event camera data, an infinite-depth alignment process is employed, which involves undistorting, rotating, and re-distorting the RGB images. This alignment ensures that the event data and RGB frames are temporally and spatially synchronized.

In our experiments, we utilize **three** comprehensive versions of DSEC tailored for object detection: *DSEC-Det* [16], *DSEC-Detection* [50], and *DSEC-MOD* [66]. A summary of the key statistics of these datasets is listed in Tab. 7.

- **DSEC-Det** [16]: This version is developed by the original DSEC team and includes annotations generated using the QDTrack multi-object tracker [11, 38]. The annotation process combines automated multi-object tracking with manual refinement to ensure high-quality

Table 7: Summary of **key statistics** from the three event datasets [16, 50, 66] used in this work.

| Dataset | Classes | Frames | Sequences | Class Names |
|---------|---------|--------|-----------|-------------|
| DSEC-MOD [66] | 1 | 13, 314 | 16 | Car |
| DSEC-Detection [50] | 3 | 52, 727 | 41 | Car
Pedestrian
Large-Vehicle |
| DSEC-Det [16] | 8 | 78, 344 | 60 | Car
Pedestrian
Bus
Bicycle
Truck
Motorcycle
Rider
Train |

and accurate detection labels. Compared to the original DSEC dataset, DSEC-Det [16] introduces additional sequences specifically designed to capture complex and dynamic urban environments, featuring crowded pedestrian areas, moving vehicles, and diverse lighting conditions [29]. These challenging scenarios provide a rich testing ground for evaluating object detection algorithms in real-world driving settings. DSEC-Det [16] comprises 60 sequences and 78,344 frames, making it the most extensive dataset used in this study. It captures diverse urban scenes with dynamic elements, such as crowded pedestrian areas and moving vehicles. Covering eight object categories relevant to autonomous driving, Car, Pedestrian, Bus, Bicycle, Truck, Motorcycle, Rider, and Train, the dataset provides a robust foundation for training and evaluating object detection models across various driving scenarios. In our experiments on DSEC-Det [16], to maintain consistency with the experimental setup of previous work DAGr [16], we report results on two categories: Car and Pedestrian.

- **DSEC-Detection** [50]: This dataset comprises 41 sequences with a total of 52,727 frames. Focusing on three fundamental object categories – Car, Pedestrian, and Large Vehicle – this version emphasizes high-precision annotations for these critical classes in autonomous driving. The initial annotations are generated using the YOLOv5 model [26] on RGB frames, leveraging its robust performance in real-time object detection. These annotations are then transferred to the corresponding event frames through homographic transformation, ensuring spatial alignment between the two modalities. A subsequent manual refinement process corrects any discrepancies and enhances annotation quality, resulting in a dataset that provides accurate and reliable labels for event-based object detection.

- **DSEC-MOD** [66]: This dataset extends the object detection capabilities to moving-object detection across diverse urban environments. It includes 16 sequences containing 13,314 frames and is specifically focused on the Car category, making it highly suitable for complex detection tasks in varied urban settings, such as intersections, highways, and residential areas. Featuring high-frequency and dense annotations, the dataset provides a valuable resource for evaluating the performance of event-based object detectors under challenging real-world conditions.

These three versions of the DSEC dataset together provide a comprehensive platform for benchmarking and evaluating event-based object detection methods, capturing a wide spectrum of scenarios, object categories, and environmental conditions. Among them, DSEC-Det [16] is the largest, most recent, and most comprehensive, annotated, and released by the original DSEC authors. Thus, we prioritize it as the primary benchmark for reporting results, ensuring relevance and reliability. DSEC-Detection [50] and DSEC-MOD [66] are datasets used by two recent event-frame fusion methods, CAFR [3] and RENet [66]. To validate the effectiveness of our method, we also report results on these two datasets.

## A.2 Baselines

To evaluate the effectiveness of our method, we compare it against both event-only and event-frame fusion state-of-the-art methods.

- **Event-Only Methods.** We include state-of-the-art event-only object detectors, namely RVT [18], SAST [39], LEOD [55], and SSM [70], which are originally trained on event-only datasets like Gen1 [10] and 1Mpx [40]. To ensure a fair comparison, we retrain these methods on the DSEC-Det [16] dataset following their respective training protocols.

- **Event-Frame Fusion Methods.** For event-frame fusion methods on DSEC-Det [16], we include DAGr [16] and HDI-Former [31], as they have been evaluated on this dataset. We report the scores of DAGr and HDI-Former from the original paper to ensure consistency and fairness. For the DSEC-Detection [50] and DSEC-MOD [66] datasets, we train our model following the standard training and evaluation settings, and compare it against state-of-the-art methods CAFR [3] and RENet [66], as reported in their respective papers. For other methods evaluated on DSEC-Detection [50] and DSEC-MOD [66], we reference the results reported in the CAFR and RENet papers, respectively.

These comparisons among state-of-the-art works ensure a fair and comprehensive evaluation while adhering to resource and code availability constraints.

## A.3 Evaluation Metrics

In this work, we adopt the mean Average Precision (**mAP**) as the primary metric to evaluate the performance of our object detection models, consistent with standard practices in the field. The mAP metric provides a comprehensive measure of detection accuracy across multiple categories and intersection-over-union (IoU) thresholds.

Mathematically, the **Average Precision (AP)** for a single class is calculated as:

$$\text{AP} = \int_0^1 p(r)\, dr \ , \tag{10}$$

where $p(r)$ represents the precision at a given recall level $r$. The mean Average Precision (mAP) is then computed as the mean of the AP values across all object categories and a range of IoU thresholds (typically from $0.5$ to $0.95$ with a step size of $0.05$). This provides an overall measure of model performance across different levels of localization precision.

In addition to mAP, we also report the following metrics from the COCO evaluation protocol [35]:

- **$\text{AP}_{50}$**: The average precision when evaluated at a fixed IoU threshold of $0.50$, indicating how well the model performs with relatively lenient localization criteria.

- **$\text{AP}_{75}$**: The average precision at a fixed IoU threshold of $0.75$, representing performance under stricter localization requirements.

- **$\text{AP}_S$, $\text{AP}_M$, and $\text{AP}_L$**: These metrics represent the average precision for small ($S$), medium ($M$), and large ($L$) objects, respectively. Object sizes are defined based on their pixel area, with **$\text{AP}_S$** typically representing objects with areas less than $32 \times 32$ pixels, **$\text{AP}_M$** representing areas between $32 \times 32$ and $96 \times 96$ pixels, and **$\text{AP}_L$** for objects larger than $96 \times 96$ pixels.

By reporting these metrics, we obtain a more nuanced understanding of the model's detection capabilities across varying object sizes and localization precision levels, ensuring a comprehensive evaluation of detection performance.

## A.4 Training & Inference Details

We train our models using mixed precision to optimize both memory efficiency and training speed. The training process spans 100,000 iterations, utilizing the Adam optimizer [27] with a OneCycle learning rate schedule [44], which gradually decays from a peak learning rate to enhance convergence.

Consistent with RVT [18], we employ a mixed batching strategy to balance computational efficiency and memory usage. Specifically:

- Standard Backpropagation Through Time (BPTT): Applied to half of the training samples, allowing for full sequence training.

Table 8: The complete results of the efficiency analysis of state-of-the-art event detectors on the test set of *DSEC-Det* [16], comparing both event-only and event-frame fusion methods. This table reports **inference times** at various frequencies, measured in **milliseconds (ms)**.

| Modal | Method | Param (M) | Operational Frequency | | | | | | |
|---|---|---|---|---|---|---|---|---|---|
| | | | 20.0 Hz | 36 Hz | 45 Hz | 60 Hz | 90 Hz | 180 Hz | 200 Hz |
| E | RVT [18] | 18.5 | 9.20 ms | 7.93 ms | 7.61 ms | 7.51 ms | 7.19 ms | 6.77 ms | 6.34 ms |
| | SAST [39] | 18.9 | 14.06 ms | 12.37 ms | 11.95 ms | 11.63 ms | 11.52 ms | 11.10 ms | 10.36 ms |
| | SSM [70] | 18.2 | 8.79 ms | 7.71 ms | 7.55 ms | 7.30 ms | 6.90 ms | 6.54 ms | 6.12 ms |
| E + F | DAGr-50 [16] | 34.6 | 73.35 ms | 55.11 ms | 51.00 ms | 48.00 ms | 45.29 ms | 43.89 ms | 37.58 ms |
| | FlexEvent | 45.4 | 14.27 ms | 13.00 ms | 12.79 ms | 12.58 ms | 12.47 ms | 12.37 ms | 12.12 ms |

- Truncated BPTT (TBPTT): Used for the other half, reducing memory usage by splitting sequences into smaller segments.

For data augmentation, we apply random horizontal flipping and zoom transformations (both zoom-in and zoom-out) to enhance the diversity of training samples.

Our training process utilizes the YOLOX framework [63], a versatile object detection framework known for its efficient and high-performing architecture. We employ a multi-component loss function to optimize our model effectively:

- Intersection over Union (IoU) Loss: This loss component measures the overlap between the predicted bounding boxes and the ground-truth boxes, ensuring that the predicted regions closely match the actual object locations.
- Classification Loss: This component evaluates the accuracy of class predictions for each detected object, ensuring that the model correctly identifies the category of each detected instance.
- Regression Loss: This loss assesses the precision of the predicted bounding box coordinates, helping the model refine the location and size of bounding boxes to align closely with the ground-truth annotations.

To ensure stable training, these loss components are averaged across both the batch and sequence length at each optimization step. This averaging process helps to reduce variance during training and facilitates smoother convergence of the model parameters. We also include a regularization term to balance the contribution of both modalities.

**Training Configuration.** The training is conducted with a batch size of 8, which provides an optimal balance between efficient GPU utilization and memory requirements. Each training sample contains a sequence length of 11 frames, allowing the model to learn temporal dependencies effectively. The frame backbone's weights are initialized using pre-trained ResNet, and the event backbone's weights are initialized using pre-trained RVT. The learning rate is set to $1 \times 10^{-4}$, following a OneCycle learning rate schedule that allows efficient exploration of the learning space and helps in achieving faster convergence.

**Hardware & Training Time.** All training experiments are carried out on two NVIDIA RTX A5000 GPUs, each with 24GB of memory, providing the computational resources necessary for handling the high-resolution event data and RGB frames. The complete training process, including all iterations and model optimization, takes approximately one day, demonstrating the efficiency of our implementation in terms of both training speed and resource utilization.

# B  Additional Quantitative Results

## B.1  Complete Results of Efficiency Analysis

We include the complete results of the efficiency analysis in Tab. 8.

## B.2  Complete Results of Ablation Study

We include the complete results of the ablation study in Tab. 9.

Table 9: The complete ablation results of different components in the FlexEvent framework. **FlexFuse** denotes the adaptive event-frame fusion module. **FlexTune** denotes the FlexTune learning mechanism. The reported results are the **mAP**, **AP$_{50}$**, **AP$_{75}$**, **AP$_S$**, **AP$_M$**, and **AP$_L$** scores on the test set of *DSEC-Det* [16]. The symbol ♦ denotes the use of interpolated ground truth labels at high frequencies in **FlexTune**. The **best** and 2nd best scores of each metric are highlighted in **bold** and underline.

| Modality | FlexTune | FlexFuse | Frequency (Hz) | mAP | AP$_{50}$ | AP$_{75}$ | AP$_S$ | AP$_M$ | AP$_L$ |
|---|---|---|---|---|---|---|---|---|---|
| | ✗ | ✗ | 20.0 | 53.2% | **77.2%** | 58.1% | **46.4%** | 64.4% | 83.0% |
| | ✗ | ✗ | 27.5 | **54.0%** | 76.8% | **59.3%** | **46.4%** | 66.6% | **85.2%** |
| | ✗ | ✗ | 30.0 | 53.5% | 75.5% | 59.3% | 45.6% | **66.8%** | 85.0% |
| Event | ✗ | ✗ | 36.0 | 52.0% | 73.3% | 58.1% | 44.0% | 65.5% | 84.9% |
| | ✗ | ✗ | 45.0 | 49.4% | 69.5% | 55.4% | 40.7% | 64.1% | 84.3% |
| | ✗ | ✗ | 60.0 | 45.9% | 64.2% | 51.8% | 36.5% | 62.3% | 82.7% |
| | ✗ | ✗ | 90.0 | 38.8% | 55.4% | 43.9% | 28.5% | 55.3% | 79.9% |
| | ✗ | ✗ | 180.0 | 22.9% | 36.1% | 23.9% | 14.1% | 34.5% | 60.1% |
| | ✓ | ✗ | 20.0 | 54.6% | **79.1%** | 61.8% | 47.4% | 64.4% | 81.4% |
| | ✓ | ✗ | 27.5 | **54.9%** | 78.8% | 61.4% | **47.6%** | 66.1% | 83.2% |
| | ✓ | ✗ | 30.0 | **54.9%** | 78.5% | 61.3% | 47.4% | **66.9%** | 83.3% |
| Event | ✓ | ✗ | 36.0 | 54.3% | 77.1% | 60.5% | 46.8% | 66.7% | 83.4% |
| | ✓ | ✗ | 45.0 | 53.3% | 75.3% | 59.8% | 45.6% | 65.4% | **83.8%** |
| | ✓ | ✗ | 60.0 | 50.7% | 72.4% | 57.3% | 42.3% | 63.5% | 83.5% |
| | ✓ | ✗ | 90.0 | 44.6% | 65.1% | 49.9% | 35.3% | 58.9% | 81.9% |
| | ✓ | ✗ | 180.0 | 30.4% | 48.1% | 32.2% | 20.7% | 44.0% | 72.9% |
| | ♦ | ✓ | 20.0 | 54.9% | 74.0% | 63.2% | 50.7% | 61.3% | 85.5% |
| | ♦ | ✓ | 27.5 | 57.3% | 75.7% | 66.3% | **52.8%** | 65.8% | 86.9% |
| | ♦ | ✓ | 30.0 | 57.7% | **75.9%** | 66.8% | 52.7% | 67.2% | **87.5%** |
| Event + Frame | ♦ | ✓ | 36.0 | **57.8%** | 75.7% | 66.5% | 52.5% | 67.9% | 87.2% |
| | ♦ | ✓ | 45.0 | 57.2% | 75.5% | 65.4% | 51.6% | **68.2%** | **87.5%** |
| | ♦ | ✓ | 60.0 | 56.1% | 74.2% | 63.4% | 50.1% | 68.1% | 86.5% |
| | ♦ | ✓ | 90.0 | 53.7% | 72.2% | 59.5% | 47.1% | 66.2% | 85.7% |
| | ♦ | ✓ | 180.0 | 48.3% | 66.9% | 52.2% | 40.8% | 60.6% | 84.2% |
| | ✗ | ✓ | 20.0 | 58.0% | 76.5% | 66.4% | 52.7% | 66.2% | 86.3% |
| | ✗ | ✓ | 27.5 | 59.6% | 78.2% | **69.6%** | **54.1%** | 69.9% | **88.0%** |
| | ✗ | ✓ | 30.0 | **60.0%** | 78.1% | 69.5% | 53.7% | **71.3%** | 87.8% |
| Event + Frame | ✗ | ✓ | 36.0 | 59.6% | 77.2% | 68.6% | 53.1% | 71.1% | 87.7% |
| | ✗ | ✓ | 45.0 | 59.0% | 76.7% | 67.1% | 52.1% | 71.1% | 87.8% |
| | ✗ | ✓ | 60.0 | 57.6% | 75.2% | 65.6% | 50.2% | 70.6% | 87.2% |
| | ✗ | ✓ | 90.0 | 54.8% | 72.6% | 61.9% | 46.8% | 68.8% | 86.3% |
| | ✗ | ✓ | 180.0 | 49.2% | 67.4% | 53.5% | 40.8% | 62.3% | 85.4% |
| | ✓ | ✓ | 20.0 | 57.4% | 78.2% | 66.6% | 51.7% | 64.9% | 83.7% |
| | ✓ | ✓ | 27.5 | 60.0% | 79.4% | 70.1% | 53.5% | 68.4% | 86.1% |
| | ✓ | ✓ | 30.0 | 60.0% | **79.7%** | **70.8%** | **53.6%** | 69.9% | 86.1% |
| Event + Frame | ✓ | ✓ | 36.0 | **60.1%** | 79.6% | **70.8%** | 53.2% | 70.3% | 85.7% |
| | ✓ | ✓ | 45.0 | 59.5% | 79.0% | 69.5% | 52.5% | 70.8% | 85.3% |
| | ✓ | ✓ | 60.0 | 58.8% | 78.5% | 69.0% | 51.1% | **71.1%** | 85.3% |
| | ✓ | ✓ | 90.0 | 56.5% | 76.5% | 65.4% | 48.2% | 70.1% | 83.8% |
| | ✓ | ✓ | 180.0 | 50.9% | 71.4% | 56.2% | 41.6% | 65.4% | 82.9% |

## B.3 Complete Results of Hyperparameter Search

We include the complete results of the hyperparameter searching in Tab. 10.

# C Additional Qualitative Results

## C.1 Qualitative Comparisons of Event Detectors

We present additional qualitative comparisons in Fig. 9, Fig. 10, and Fig. 11 to further illustrate the advantages of FlexEvent over three state-of-the-art methods – RVT [18], SAST [39], and DAGr [16] – across nine diverse scenes.

As shown in Fig. 9, under fast-motion conditions, RVT and SAST fail to detect pedestrians, while DAGr misclassifies many distant objects as pedestrians or vehicles. In contrast, FlexEvent accurately captures all objects, demonstrating superior robustness in challenging high-speed scenarios.

Similarly, in Fig. 10, RVT frequently misses objects in cluttered scenes, and both SAST and DAGr mistakenly recognize pedestrians and distant vehicles under noisy, occluded conditions. Here again,

Table 10: The complete ablation results of different hyperparameter configurations in the FlexEvent framework. $\tau^{car}$, $\tau^{ped}$ denotes the confidence threshold for car and pedestrian, respectively. $\tau^{iou}$ denotes the IoU threshold when filter by tracking, $\mathbf{L}^{track}$ denotes the minimum track length. The reported results are the **mAP** scores on the test set of *DSEC-Det* [16]. The **best** and 2nd best scores of each metric from each hyperparameter configuration are highlighted in **bold** and underline.

| $\tau^{car}$ | $\tau^{ped}$ | $\mathbf{L}^{track}$ | $\tau^{iou}$ | Frequency (Hz) | mAP | AP$_{50}$ | AP$_{75}$ | AP$_{S}$ | AP$_{M}$ | AP$_{L}$ |
|---|---|---|---|---|---|---|---|---|---|---|
| 0.6 | 0.3 | 10 | 0.8 | 20.0 | 56.5% | **81.3%** | 66.4% | 51.8% | 62.2% | 82.8% |
| 0.6 | 0.3 | 10 | 0.8 | 27.5 | 55.9% | 74.7% | 65.1% | 51.9% | 61.4% | 87.0% |
| 0.6 | 0.3 | 10 | 0.8 | 30.0 | 56.7% | 75.3% | 66.0% | 52.3% | 63.5% | 87.0% |
| 0.6 | 0.3 | 10 | 0.8 | 36.0 | **57.2%** | 75.9% | **67.0%** | **52.5%** | 64.8% | 86.9% |
| 0.6 | 0.3 | 10 | 0.8 | 45.0 | 57.1% | 75.7% | 66.2% | 51.5% | 66.1% | 87.2% |
| 0.6 | 0.3 | 10 | 0.8 | 60.0 | 56.7% | 75.3% | 65.7% | 50.3% | **67.4%** | **87.3%** |
| 0.6 | 0.3 | 10 | 0.8 | 90.0 | 54.5% | 73.2% | 62.3% | 47.3% | 66.3% | 86.2% |
| 0.6 | 0.3 | 10 | 0.8 | 180.0 | 49.2% | 68.2% | 54.2% | 40.8% | 62.4% | 85.5% |
| 0.6 | 0.3 | 10 | 0.6 | 20.0 | 56.7% | **80.6%** | 65.5% | 51.2% | 63.0% | 81.7% |
| 0.6 | 0.3 | 10 | 0.6 | 27.5 | 57.2% | 79.3% | 65.0% | 51.9% | 65.2% | 84.5% |
| 0.6 | 0.3 | 10 | 0.6 | 30.0 | 57.7% | 79.4% | 66.0% | **52.3%** | 66.3% | 85.0% |
| 0.6 | 0.3 | 10 | 0.6 | 36.0 | **57.9%** | 79.5% | **66.4%** | 52.2% | 66.8% | 84.8% |
| 0.6 | 0.3 | 10 | 0.6 | 45.0 | 57.7% | 79.2% | 65.6% | 51.7% | 67.1% | 85.1% |
| 0.6 | 0.3 | 10 | 0.6 | 60.0 | 57.0% | 78.8% | 64.8% | 50.5% | **67.6%** | **85.3%** |
| 0.6 | 0.3 | 10 | 0.6 | 90.0 | 54.3% | 76.4% | 60.0% | 46.9% | 66.3% | 84.5% |
| 0.6 | 0.3 | 10 | 0.6 | 180.0 | 47.0% | 69.0% | 49.1% | 37.8% | 61.1% | 83.9% |
| 0.6 | 0.3 | 8 | 0.6 | 20.0 | 56.3% | 77.2% | 64.9% | 50.4% | 64.1% | 83.3% |
| 0.6 | 0.3 | 8 | 0.6 | 27.5 | 58.5% | 78.3% | 68.1% | 52.5% | 66.8% | 84.8% |
| 0.6 | 0.3 | 8 | 0.6 | 30.0 | 58.8% | 78.5% | 68.7% | 52.6% | 67.9% | 85.7% |
| 0.6 | 0.3 | 8 | 0.6 | 36.0 | **59.1%** | **78.8%** | **69.0%** | **52.7%** | 68.8% | **86.5%** |
| 0.6 | 0.3 | 8 | 0.6 | 45.0 | 58.8% | 78.2% | 68.6% | 52.3% | 68.8% | 86.1% |
| 0.6 | 0.3 | 8 | 0.6 | 60.0 | 58.4% | 77.9% | 67.5% | 51.3% | **69.6%** | 85.5% |
| 0.6 | 0.3 | 8 | 0.6 | 90.0 | 56.2% | 76.6% | 64.8% | 48.5% | 68.3% | 84.7% |
| 0.6 | 0.3 | 8 | 0.6 | 180.0 | 51.2% | 71.9% | 56.3% | 42.6% | 64.2% | 82.9% |
| 0.6 | 0.3 | 6 | 0.6 | 20.0 | 57.3% | 80.0% | 65.2% | 51.2% | 65.8% | 84.1% |
| 0.6 | 0.3 | 6 | 0.6 | 27.5 | 59.4% | 81.3% | 68.5% | 53.4% | 68.8% | **85.7%** |
| 0.6 | 0.3 | 6 | 0.6 | 30.0 | 59.7% | **81.7%** | 69.0% | **53.7%** | 69.0% | 85.3% |
| 0.6 | 0.3 | 6 | 0.6 | 36.0 | **59.9%** | 81.4% | 69.0% | 53.6% | **69.8%** | **85.7%** |
| 0.6 | 0.3 | 6 | 0.6 | 45.0 | 59.3% | 80.5% | 67.9% | 52.8% | 69.4% | 85.6% |
| 0.6 | 0.3 | 6 | 0.6 | 60.0 | 58.5% | 79.6% | 67.0% | 51.5% | 69.6% | 84.9% |
| 0.6 | 0.3 | 6 | 0.6 | 90.0 | 55.7% | 77.2% | 62.8% | 48.1% | 67.9% | 84.3% |
| 0.6 | 0.3 | 6 | 0.6 | 180.0 | 48.8% | 70.6% | 50.9% | 40.8% | 61.1% | 83.4% |
| 0.6 | 0.6 | 6 | 0.6 | 20.0 | 57.4% | 78.2% | 66.6% | 51.7% | 64.9% | 83.7% |
| 0.6 | 0.6 | 6 | 0.6 | 27.5 | 60.0% | 79.4% | 70.1% | 53.5% | 68.4% | **86.1%** |
| 0.6 | 0.6 | 6 | 0.6 | 30.0 | 60.0% | **79.7%** | **70.8%** | **53.6%** | 69.9% | **86.1%** |
| 0.6 | 0.6 | 6 | 0.6 | 36.0 | **60.1%** | 79.6% | **70.8%** | 53.2% | 70.3% | 85.7% |
| 0.6 | 0.6 | 6 | 0.6 | 45.0 | 59.5% | 79.0% | 69.5% | 52.5% | 70.8% | 85.3% |
| 0.6 | 0.6 | 6 | 0.6 | 60.0 | 58.8% | 78.5% | 69.0% | 51.1% | **71.1%** | 85.3% |
| 0.6 | 0.6 | 6 | 0.6 | 90.0 | 56.5% | 76.5% | 65.4% | 48.2% | 70.1% | 83.8% |
| 0.6 | 0.6 | 6 | 0.6 | 180.0 | 50.9% | 71.4% | 56.2% | 41.6% | 65.4% | 82.9% |
| 0.8 | 0.8 | 6 | 0.6 | 20.0 | 56.6% | 80.7% | 65.5% | 50.8% | 65.4% | 82.6% |
| 0.8 | 0.8 | 6 | 0.6 | 27.5 | 58.7% | 81.9% | 68.9% | **52.8%** | 68.6% | 84.9% |
| 0.8 | 0.8 | 6 | 0.6 | 30.0 | **59.1%** | **82.0%** | **69.2%** | 52.7% | 69.5% | 84.8% |
| 0.8 | 0.8 | 6 | 0.6 | 36.0 | 58.9% | 81.7% | 68.8% | 52.6% | **69.9%** | 85.0% |
| 0.8 | 0.8 | 6 | 0.6 | 45.0 | 58.4% | 81.4% | 67.7% | 51.6% | 69.8% | **85.1%** |
| 0.8 | 0.8 | 6 | 0.6 | 60.0 | 57.4% | 80.1% | 66.6% | 50.2% | **69.9%** | 84.3% |
| 0.8 | 0.8 | 6 | 0.6 | 90.0 | 55.7% | 78.6% | 63.7% | 47.8% | 68.9% | 84.2% |
| 0.8 | 0.8 | 6 | 0.6 | 180.0 | 50.2% | 74.0% | 55.2% | 41.5% | 63.7% | 83.1% |

FlexEvent excels by reliably detecting all relevant objects, highlighting its ability to handle complex urban environments.

Finally, in Fig. 11, rapidly changing illumination leads RVT and DAGr to misinterpret subtle motion cues; yet FlexEvent preserves stable performance, underscoring its enhanced temporal awareness. These observations confirm that FlexEvent not only surpasses competing methods in quantitative benchmarks but also consistently delivers high detection accuracy in real-world conditions – ensuring reliable performance in fast-moving, cluttered, or dynamically lit scenes.

## C.2  Comparisons under Different Frequencies

We further evaluate the performance of FlexEvent across diverse event frequencies in Fig. 12, Fig. 13, and Fig. 14. Specifically, we compare our method with RVT under settings ranging from low-frequency (20 Hz) to high-frequency (180 Hz) event streams, including an extreme scenario with empty events.

In Fig. 12, RVT struggles to detect distant or partially occluded objects at both low and high frequencies, whereas FlexEvent consistently identifies all targets by adaptively fusing frame-based evidence with event cues.

Similarly, in Fig. 13, RVT suffers substantial performance drops under sparse event conditions, often overlooking pedestrians; yet FlexEvent maintains reliable detection through its frequency-adaptive training strategy.

Finally, in Fig. 14, even when event input is minimal or missing, FlexEvent retains high accuracy by leveraging complementary frame information. These comparisons reaffirm our robustness and adaptability, highlighting our ability to handle a broad spectrum of event frequencies while preserving superior detection performance.

# D  Potential Societal Impact & Limitations

In this section, we discuss the potential societal impact of FlexEvent, including its positive contributions, broader implications, and known limitations. While our method offers significant advancements in event camera object detection, it is important to consider its broader consequences and areas for future improvement.

## D.1  Societal Impact

The development of FlexEvent introduces several positive societal benefits, particularly in safety-critical applications such as autonomous driving, robotics, and surveillance. By enhancing the ability to detect fast-moving objects in real time, our framework can improve the responsiveness and safety of autonomous systems operating in dynamic environments. This is especially important for avoiding collisions or responding to hazards in high-speed scenarios. For example, autonomous vehicles equipped with our approach can better detect pedestrians, cyclists, and other vehicles in real time, potentially reducing accidents and saving lives.

Additionally, the computational efficiency provided by the adaptive event-frame fusion (FlexFuse) and frequency-adaptive learning (FlexTune) mechanisms reduces the need for resource-intensive training processes. This contributes to the broader societal goal of making advanced AI technologies more accessible and less energy-intensive, thereby minimizing the environmental impact of large-scale AI models. Our approach could also benefit industries beyond transportation, such as robotics for healthcare, industrial automation, and public safety.

## D.2  Broader Impact

The broader implications of FlexEvent include its potential to advance the field of event-based vision and enable new applications where high temporal resolution is crucial. By overcoming the limitations of conventional fixed-frequency object detection methods, our approach paves the way for more flexible, adaptable AI systems. This could lead to improvements in areas such as drone navigation, real-time video analysis for security purposes, and human-robot collaboration, where detecting fast-moving objects and adapting to changing environments are critical.

Moreover, the development of efficient and scalable detection systems like our approach can drive further innovation in resource-constrained environments, such as low-power edge devices. These advancements could make high-performance detection systems more widely available, particularly in developing regions or areas with limited access to computational resources.

However, as with any powerful technology, there is a risk of misuse. Enhanced object detection capabilities could potentially be exploited for surveillance purposes, raising privacy concerns. As event camera technology becomes more widespread, it is important to establish ethical guidelines

and regulatory frameworks to ensure that these systems are used responsibly, particularly when monitoring public spaces or collecting sensitive data.

### D.3 Known Limitations

While FlexEvent demonstrates significant performance improvements, there are several known limitations to our approach, which can be summarized as follows:

**Dependence on the Quality of Event Camera Data.** The effectiveness of our approach relies on the quality of the event camera sensor. Inconsistent or noisy event data, especially under poor lighting or extreme weather conditions, could affect detection performance. Future work could explore robustness to sensor noise and adaptation to diverse environmental conditions.

**Limited Generalization to Unseen Scenarios.** Although our approach shows strong performance across varying frequencies, it may still face challenges in completely unseen environments, where the motion dynamics and scene conditions differ significantly from the training data. Investigating methods for domain adaptation or online learning could help improve generalization to new contexts.

**Resource Requirements for High-Frequency Data.** While FlexFuse mitigates the computational cost of training on high-frequency event data, processing extremely high-frequency event streams still requires substantial computational resources during inference. This could limit the scalability on resource-constrained devices or in real-time applications with stringent latency requirements.

# E    Public Resources Used

In this section, we acknowledge the public resources used, during the course of this work.

## E.1    Public Datasets Used

- DSEC[2] . . . . . . . . . . . . . . . . . . . . . . . . . . . . . . . . . . . . . . . . . . . . . . . . . . . . . . . . . . . . . . . . CC BY-SA 4.0
- DSEC-Det[3] . . . . . . . . . . . . . . . . . . . . . . . . . . . . . . . . . . . . . . . GNU General Public License v3.0
- DSEC-Detection[4] . . . . . . . . . . . . . . . . . . . . . . . . . . . . Creative Commons Zero v1.0 Universal
- DSEC-MOD[5] . . . . . . . . . . . . . . . . . . . . . . . . . . . . . . . . . . . . . . . . . . . . . . . . . . . . . . . . . Unknown
- Gen 1[6] . . . . . . . . . . . . . . . . . . . . . . . . Prophesee Gen1 Automotive Detection Dataset License
- 1 Mpx[7] . . . . . . . . . . . . . . . . . . . . . . . Prophesee 1Mpx Automotive Detection Dataset License

## E.2    Public Implementations Used

- RVT[8] . . . . . . . . . . . . . . . . . . . . . . . . . . . . . . . . . . . . . . . . . . . . . . . . . . . . . . . . . . . . . . MIT License
- SAST[9] . . . . . . . . . . . . . . . . . . . . . . . . . . . . . . . . . . . . . . . . . . . . . . . . . . . . . . . . . . . . MIT License
- SSM[10] . . . . . . . . . . . . . . . . . . . . . . . . . . . . . . . . . . . . . . . . . . . . . . . . . . . . . . . . . . . Unknown
- LEOD[11] . . . . . . . . . . . . . . . . . . . . . . . . . . . . . . . . . . . . . . . . . . . . . . . . . . . . . . . . . MIT License
- DAGr[12] . . . . . . . . . . . . . . . . . . . . . . . . . . . . . . . . . . . . . . . GNU General Public License v3.0
- RENet[13] . . . . . . . . . . . . . . . . . . . . . . . . . . . . . . . . . . . . . . . . . . . . . . . . . . . . . . . . . . . Unknown

---

[2] https://dsec.ifi.uzh.ch
[3] https://github.com/uzh-rpg/dsec-det
[4] https://github.com/abhishek1411/event-rgb-fusion
[5] https://github.com/ZZY-Zhou/RENet
[6] https://www.prophesee.ai/2020/01/24/prophesee-gen1-automotive-detection-dataset
[7] https://www.prophesee.ai/2020/11/24/automotive-megapixel-event-based-dataset
[8] https://github.com/uzh-rpg/RVT
[9] https://github.com/Peterande/SAST
[10] https://github.com/uzh-rpg/ssms_event_cameras
[11] https://github.com/Wuziyi616/LEOD
[12] https://github.com/uzh-rpg/dagr
[13] https://github.com/ZZY-Zhou/RENet

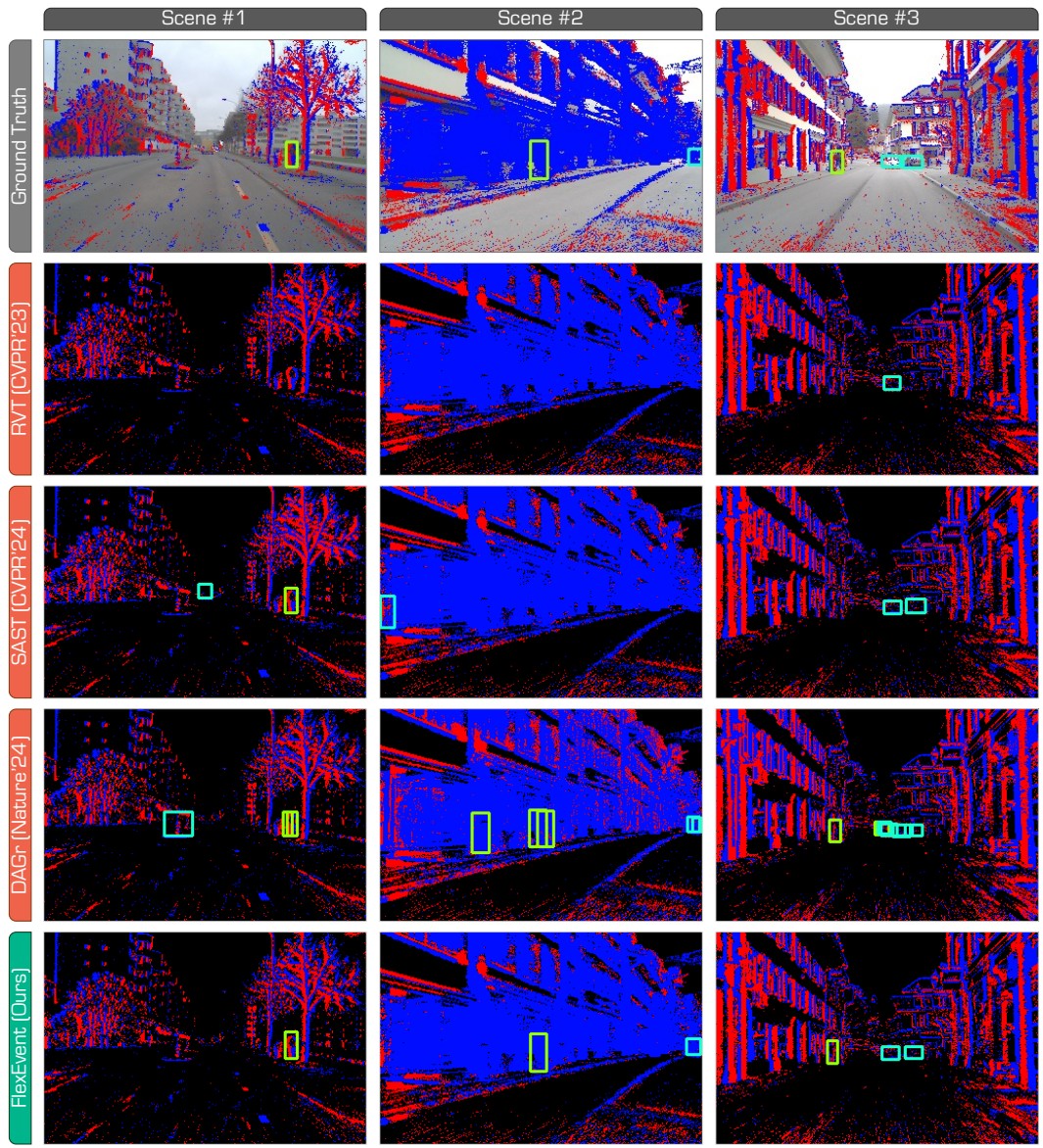

Figure 9: Additional qualitative results of state-of-the-art event camera detectors. We compare the proposed FlexEvent with RVT [18], SAST [39], and DAGr [16] on the test set of *DSEC-Det*. Best viewed in colors.

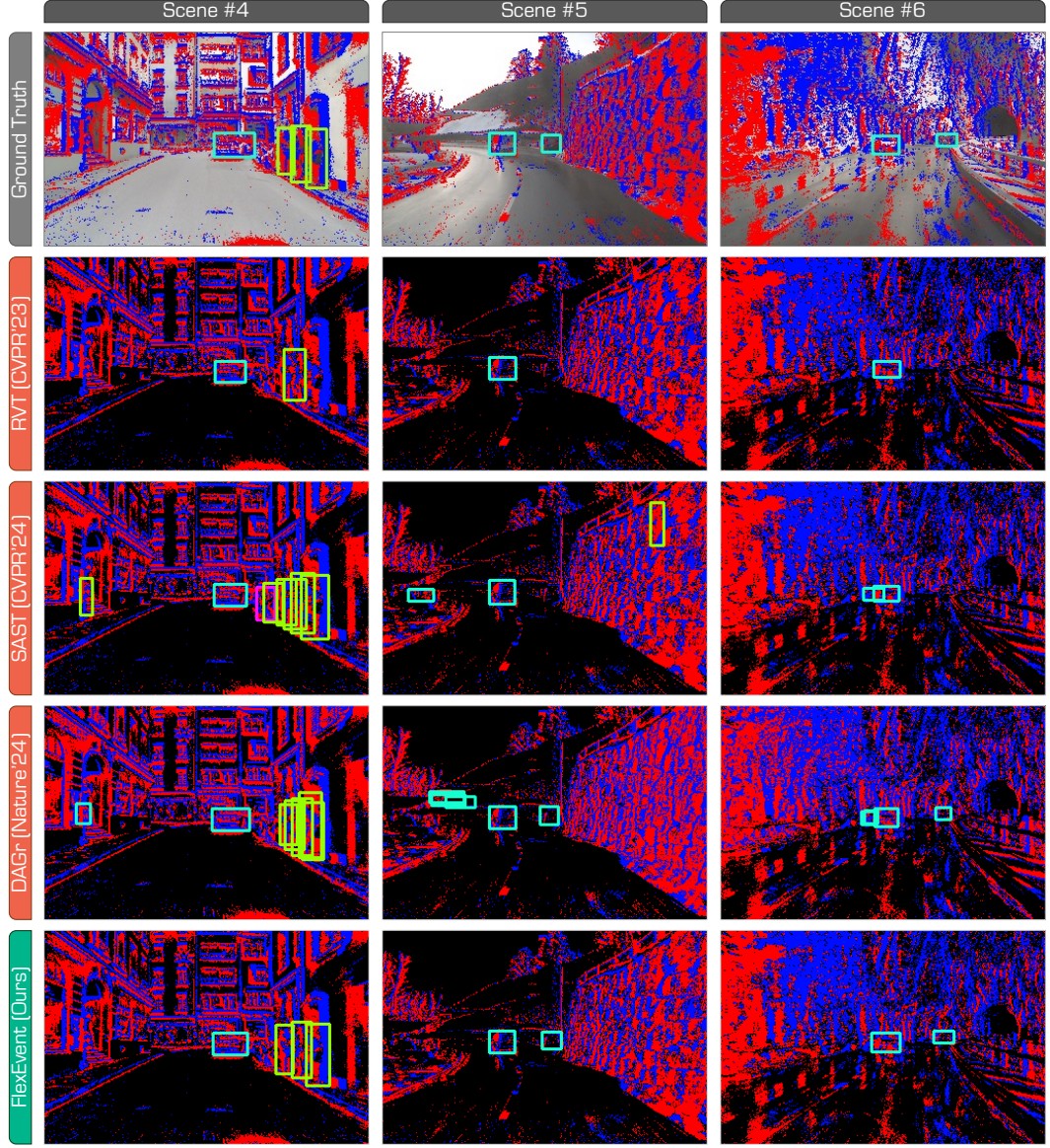

Figure 10: Additional qualitative results of state-of-the-art event camera detectors. We compare the proposed FlexEvent with RVT [18], SAST [39], and DAGr [16] on the test set of *DSEC-Det*. Best viewed in colors.

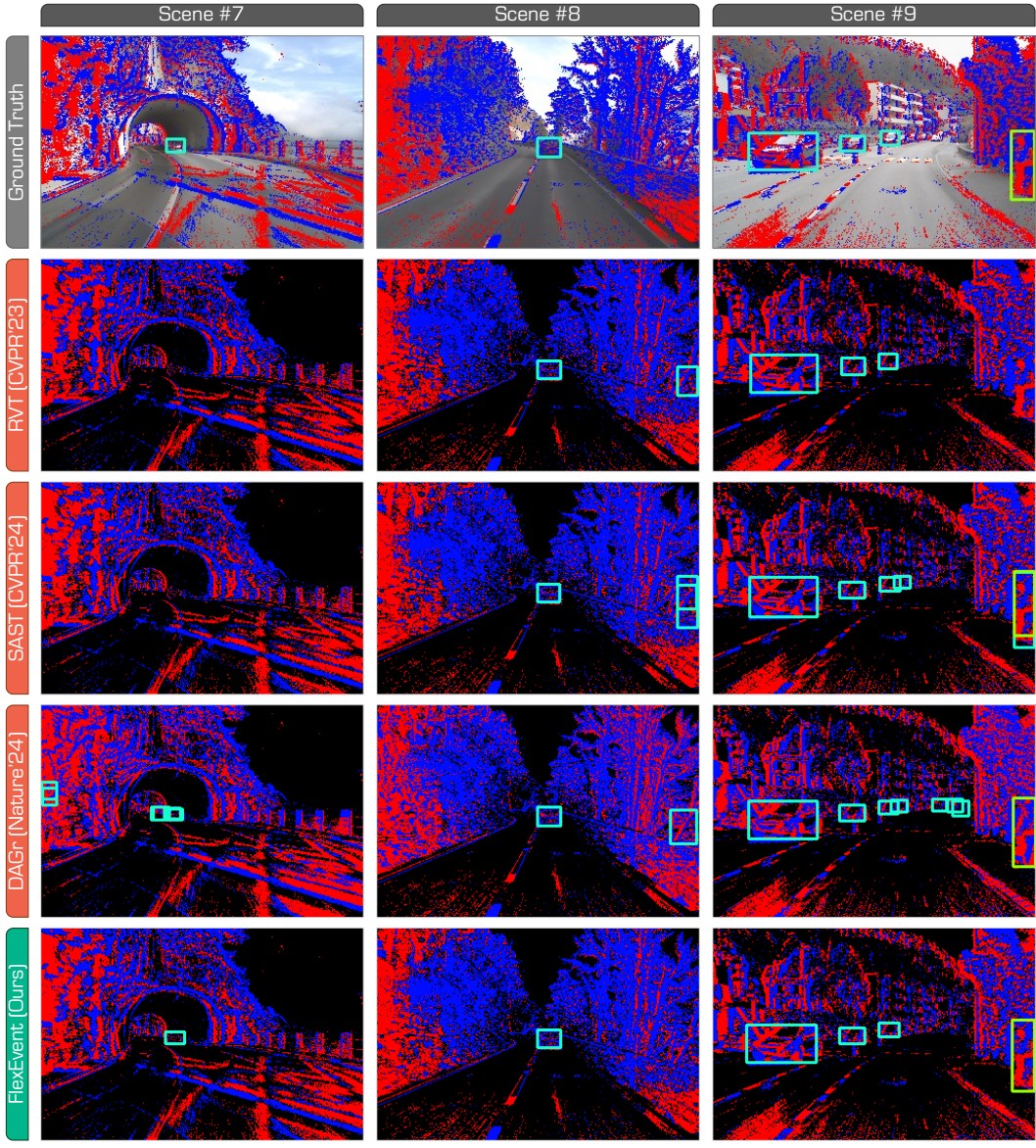

Figure 11: Additional qualitative results of state-of-the-art event camera detectors. We compare the proposed FlexEvent with RVT [18], SAST [39], and DAGr [16] on the test set of *DSEC-Det*. Best viewed in colors.

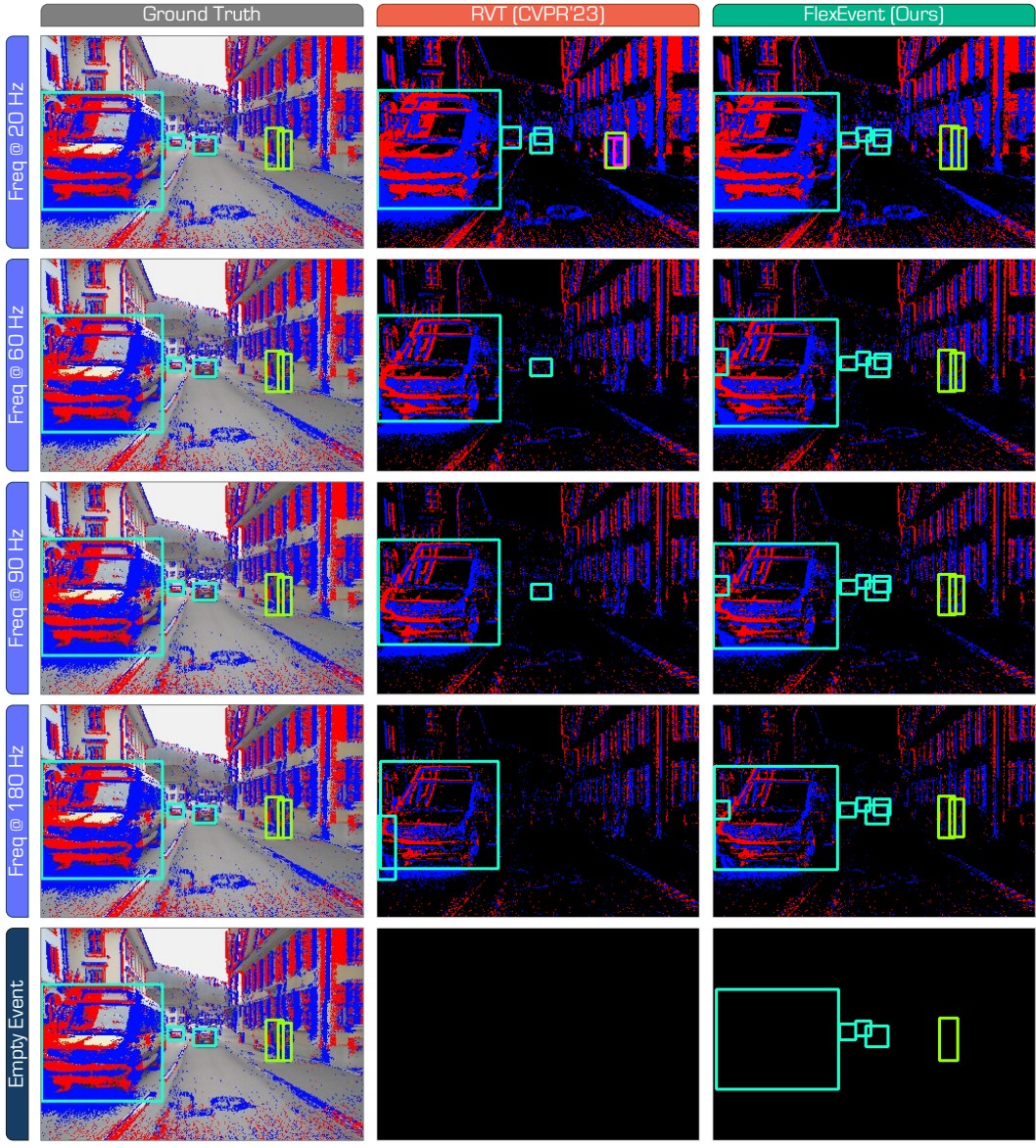

Figure 12: Additional qualitative comparisons of the RVT model [18] and the proposed FlexEvent under different event camera operation frequencies (20 Hz, 60 Hz, 90 Hz, and 180 Hz) and the empty event scenario. The experiments are conducted on the test set of *DSEC-Det*. Best viewed in colors.

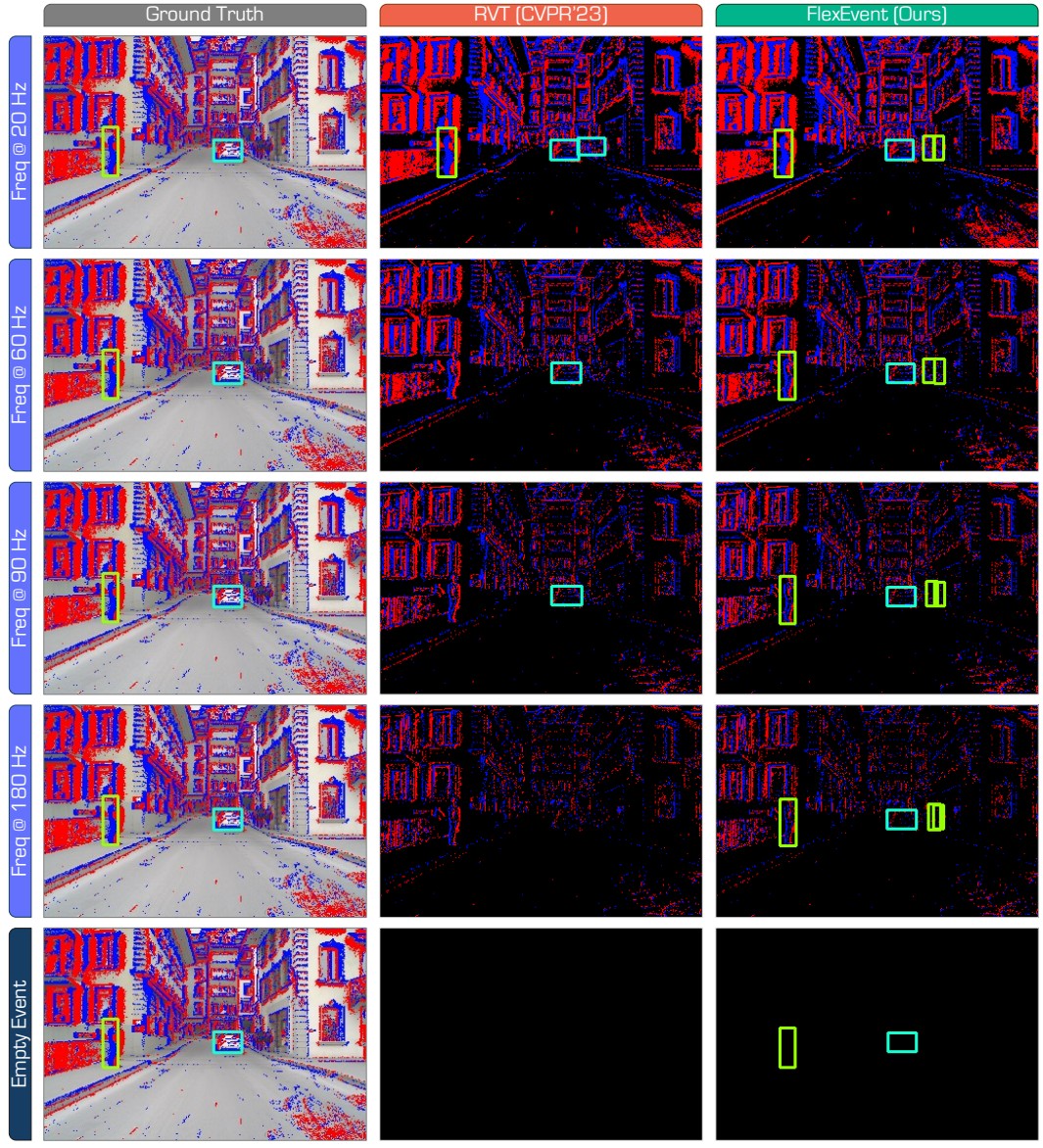

Figure 13: Additional qualitative comparisons of the RVT model [18] and the proposed FlexEvent under different event camera operation frequencies (20 Hz, 60 Hz, 90 Hz, and 180 Hz) and the empty event scenario. The experiments are conducted on the test set of *DSEC-Det*. Best viewed in colors.

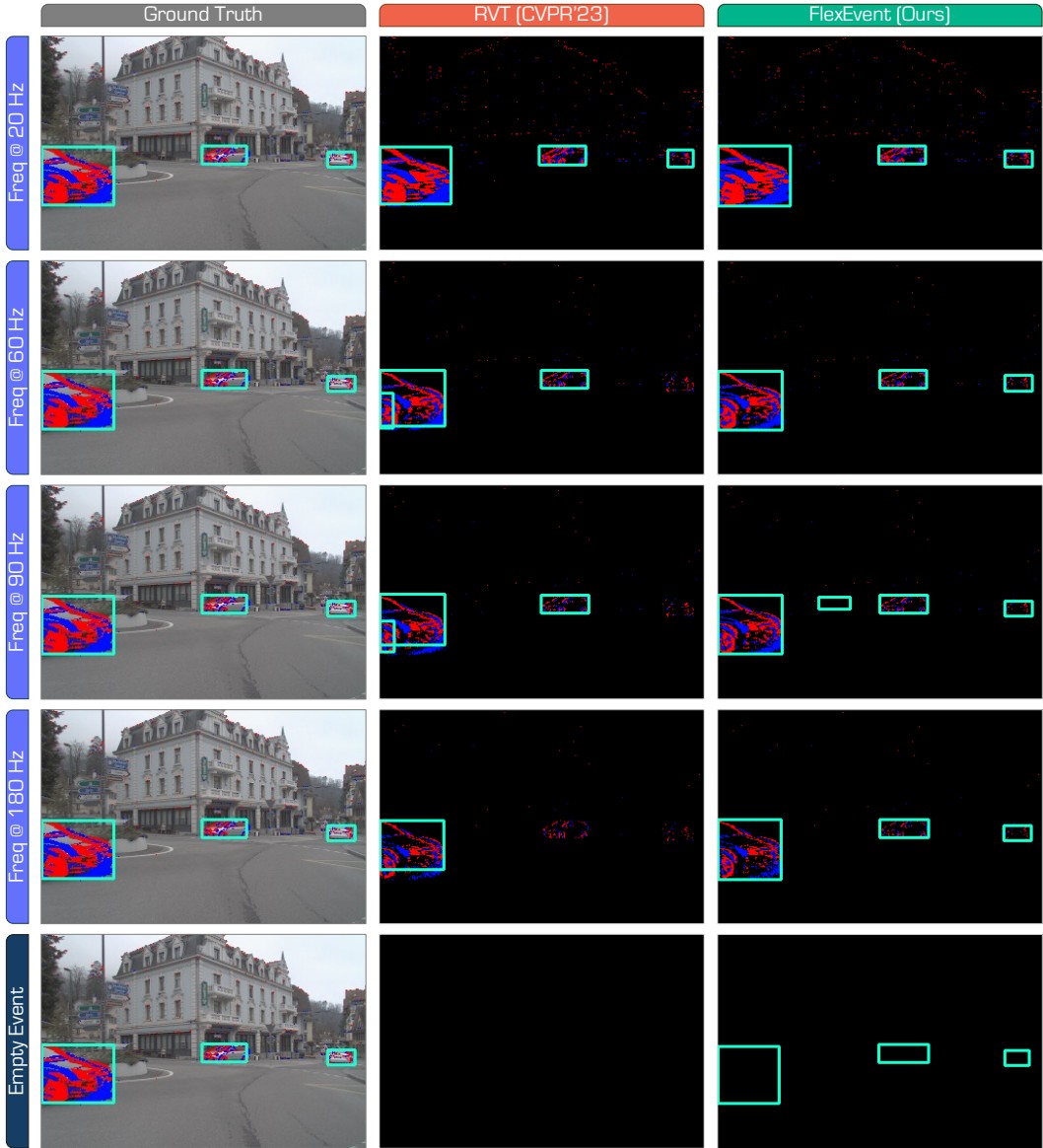

Figure 14: Additional qualitative comparisons of the RVT model [18] and the proposed FlexEvent under different event camera operation frequencies (20 Hz, 60 Hz, 90 Hz, and 180 Hz) and the empty event scenario. The experiments are conducted on the test set of *DSEC-Det*. Best viewed in colors.

