# OpenReview forum: "FlexEvent: Towards Flexible Event-Frame Object Detection at Varying Operational Frequencies"
_NeurIPS.cc/2025/Conference — NeurIPS 2025 poster_

### Official Review · Reviewer_RR3R · 2025-06-03

**Clarity:** 4
**Significance:** 4
**Originality:** 4
**Rating:** 5
**Confidence:** 3

**Summary:**

This paper proposes FlexEvent, an RGB-event fusion-based detector that adapts to varying input frequencies. It includes FlexFuse, an adaptive fusion module that integrates event and RGB data, and FlexTune, a strategy that adjusts supervision signals to improve generalization across input rates.

**Questions:**

In Section 3.1, regarding Event Processing:
- Does Eq. (2) use Kronecker delta functions? This should be clarified; otherwise, the unspecified meaning of \\( \delta \\) may puzzle readers.
- The event representation and pre-processing follow the design in RVT, but no citation is provided here. A proper reference should be added.
- The paper states that events are constructed into a 4D tensor, but it is unclear how such a 4D tensor is processed with 2D convolutions for subsequent RGB-event fusion. RVT gives a detailed explanation, and a similar clarification should be included here.

In Section 3.2, regarding FlexFuse:
- For dynamic event aggregation with \\(\Delta t_a = 100\\) ms and \\(a = 10\\), assuming \\(b = 100\\), the time interval is split into 10 ms sub-intervals, and a high-frequency event sample \\(E_b\\) is randomly selected from these sub-intervals. How does this ensure that the selected \\(E_b\\) (e.g., from 10--20 ms) is aligned with the prediction timestamp at the end of \\(\Delta t_a\\) (i.e., 100 ms)?
- RGB and event cameras may have different intrinsic/extrinsic calibrations and resolutions, and event images have object duplication due to temporal accumulation. Has the fusion method considered alignment between these two modalities?

Regarding loss function:
- In Eq. (7), what value is used for the weighting factor \\(\lambda\\)? Also, if \\(\lambda = 0\\) (i.e., no regularization is applied), how does this affect the performance?
- Similarly, in the definition of \\( \mathcal{L} _ {tune} = \mathcal{L} _ {GT} + \beta \sum \mathcal{L} _ {det}(\tilde{y}, \hat{y}) \\), what value is used for the coefficient \\(\beta\\)? Also, when the frequency is sufficiently high, the low-frequency label \\(y\\) and the high-frequency pseudo label \\(\tilde{y}\\) become very similar. I have concern about the necessity of the term \\(\mathcal{L} _ {det}(\tilde{y}, \hat{y})\\).

**Ethical Concerns:**

["NO or VERY MINOR ethics concerns only"]

**Final Justification:**

The rebuttal addresses most concerns, so I maintain my original score.

**Limitations:**

The limitations have been discussed in the appendix.

**Paper Formatting Concerns:**

No major formatting issues found.

**Quality:**

4

**Strengths And Weaknesses:**

**Strengths**
1. Good presentation and clear methodology
2. Comprehensive experiment setup
3. Code is promised to be released, which will benefit the community
4. Significant improvement over prior SOTA methods

**Weaknesses**
Some details are confusing to me. Please refer to the **Questions** below.

---

> ### Author Rebuttal · Authors · 2025-07-31
>
> We sincerely thank Reviewer `RR3R` for the thoughtful assessment and helpful suggestions, and we appreciate the recognition of our good presentation and comprehensive experiments. Below, we respond to your comments and suggestions in detail.
>
> ---
> > **`Q1`:** *"Does Eq. (2) use Kronecker delta functions? Please clarify the notation."*
>
> **A:** Thanks for your comment. Yes. In Eq. `2`, the symbols $\delta(·)$ denote **Kronecker deltas over discrete indices (polarity, time bin, and pixel coordinates)**. We will add a sentence in Sec. `3.1`, stating this explicitly to avoid ambiguity. This event-tensor construction (discrete binning with deltas) is standard and follows the RVT formulation.
>
> ---
> > **`Q2`:** *"The event representation and pre-processing follow the design in RVT, but no citation is provided here."*
>
> **A:** Thank you for asking. We apologize for our oversight. We will add an explicit citation to RVT in Sec. `3.1`, where we construct the event tensor, and state that our implementation follows RVT to enable 2D convolutions.
>
> ---
> > **`Q3`:** *"It is unclear how a 4D tensor is processed with 2D convolutions for subsequent RGB-event fusion."*
>
> **A:** Thanks for your insightful concern. As in RVT, we **flatten polarity and time into the channel axis**, converting the 4D tensor to a 3D tensor of shape (2T, H, W) that is directly compatible with 2D convolutions. We will add more details about the data processing in the supplementary material.
>
> ---
> > **`Q4`:** *"With dynamic aggregation, how is the high-frequency event slice aligned with the prediction time at the end of the sampling interval?"*
>
> **A:** We appreciate this valuable question. Our detection is **anchored at the end timestamp** $t_2$ of the labeled interval $[t_1, t_2]$ with $t_2 = t_1 + \Delta t_a$. The inputs are: the RGB frame $F(t_2)$, the low-frequency event aggregation $E_a=[t_1, t_2]$, and one high-frequency event slice $E_b$ of duration $\Delta t_b$. We randomly sample $E_b$ within $[t_1, t_2]$; in practice, we allow a **millisecond-scale jitter** so the slice may end at $t_2-\varepsilon$ ($\varepsilon$ is small), which serves as implicit temporal augmentation against synchronization noise. This keeps the prediction target (the boxes at $t_2$) **aligned** with its preceding event evidence, ensuring $(F, E_a, E_b)$ are consistently paired across modalities and frequencies for subsequent fusion and decoding.
>
> ---
> > **`Q5`:** *"RGB and event cameras may differ in intrinsics/extrinsics/resolution; events also show duplication from temporal accumulation. Did you consider cross-modal alignment?"*
>
> **A:** Yes. We **synchronize the event stream and frames** over $[t_1, t_2]$ and operate in a common image plane $H \times W$ before fusion (resize/rectify using the dataset’s provided calibration). FlexFuse then adapts the contribution of the two modalities via learned soft weights, which mitigates residual appearance/geometric mismatch and duplication artifacts at the feature level. We will add a short implementation note on using the provided calibration and the shared resolution in the revised version.
>
> ---
> > **`Q6`:** *"In Eq. (7), what is the weighting factor $\lambda$? What happens if $\lambda=0$ (no regularization)?"*
>
> **A:** Thank you for pointing this out. Eq. `7` adds a variance-over-mean regularizer on the fusion gates ($\alpha$, $\beta$) with weight $\lambda$. We use a small positive $\lambda = 0.1$ chosen on the validation set. This regularization ensures balanced utilization of both the event and frame branches,  and it penalizes large variations in the soft weights, encouraging a more uniform contribution from both modalities. Intuitively, setting $\lambda = 0$ removes the balance constraint and can **let one branch dominate**. In practice, we observe a small drop in mAP without this term, and we will report the numbers in an ablation table in the revised version.
>
> ---
> > **`Q7`:** *"What is $\beta$ in $\mathcal{L}_\mathrm{tune}$? If the low-frequency label and high-frequency pseudo label become very similar at high rates, is the term still necessary?"*
>
> **A:** Thanks for raising this point. The tuning loss mixes human GT and high-frequency pseudo labels with coefficient $\beta$ (chosen on validation, **$\beta = 0.5$** in our experiments to reflect moderate trust in the pseudo labels). When the low-frequency label $y$ and the high-frequency pseudo label $\tilde y$ become very similar at high rates, the pseudo-label term behaves as a **consistency regularizer** across dense time steps: if $\hat y \approx \tilde y \approx y$, its contribution (and gradient) is near zero, so it **does not bias** training; if small misalignments arise from sync jitter or occlusions, it **stabilizes** predictions and helps propagate supervision between sparse GT frames. In ablations, setting **$\beta=0$** means not using pseudo high-frequency label, which yields a performance drop (48 → 43.7), especially at mid/high frequencies.
>
> As a practical refinement, we can **confidence-weight** the pseudo-label term and/or **decay $\beta$** with frequency to avoid over-reliance when $\tilde y$ is already near-perfect.
>
> ---
> Last but not least, we would like to sincerely thank Reviewer `RR3R` again for the valuable time and constructive feedback provided during this review.

---

> > ### Author Response · Authors · 2025-08-03
> > **Looking forward to hearing from you**
> >
> > Dear Reviewer `RR3R`,
> >
> > Thank you very much for your thoughtful and detailed review of our submission!
> >
> > Following your suggestions, we clarified the notation, added missing citations to RVT, and elaborated on how the event tensor is processed using 2D convolutions. We also included implementation details on cross-modal alignment, clarified the fusion regularizer in Eq. (7), and expanded the discussion on label weighting in the tuning loss.
> >
> > We appreciate your valuable comments, which helped improve the clarity and rigor of the work.
> >
> > Please feel free to reach out during the Author-Reviewer Discussion phase if further clarification is needed!
> >
> > *Best regards,*
> >
> > The Authors of Submission 1326

---

> > > ### Comment · Reviewer_RR3R · 2025-08-04
> > >
> > > Thank you for your response. Most of my concerns have been resolved. I will keep my rating

---

> > > > ### Author Response · Authors · 2025-08-04
> > > > **Thank you for your acknowledgment**
> > > >
> > > > Dear Reviewer `RR3R`,
> > > >
> > > > We sincerely appreciate your acknowledgment of our rebuttal. We are glad to hear that your concerns have been addressed and are voting toward acceptance.
> > > >
> > > > Your support and encouragement mean a lot to us!
> > > >
> > > > *Best regards,*
> > > >
> > > > The Authors of Submission 1326

---

### Official Review · Reviewer_BJuF · 2025-06-25

**Clarity:** 3
**Significance:** 3
**Originality:** 2
**Rating:** 4
**Confidence:** 2

**Summary:**

The paper proposes FlexEvent, a novel framework for event-based object detection that is designed to operate effectively across varying operational frequencies. The authors evaluate FlexEvent on multiple large-scale event camera datasets (DSEC-Det, DSEC-Detection, DSEC-MOD). The proposed method outperforms both event-only and event-frame fusion baselines, showing significant improvements in mean average precision (mAP), particularly under high-frequency and challenging dynamic scenarios. The framework also demonstrates efficiency in terms of inference time and parameter count, supporting its potential for real-time applications.

**Questions:**

1）Demonstrate robustness to modality-specific noise. The adaptive fusion in FlexFuse dynamically balances event and frame features, but it’s unclear how this affects performance in scenarios where one modality is significantly noisier (e.g., low-light frames or sparse events).
2）Show FlexTune’s superiority over semi-supervised baselines. FlexTune uses pseudo-labels for high-frequency data, but the paper does not compare against other semi-supervised or self-training methods (e.g., consistency regularization, teacher-student models).
3）Prove feasibility for edge deployment. While the paper claims efficiency, the model size (45.4M parameters) is larger than event-only baselines (e.g., RVT: 18.5M). How does this impact real-world deployment on resource-constrained platforms?
4）Validate generalization to untrained frequencies. The model is evaluated on frequencies up to 180Hz, but it’s unclear how it extrapolates to intermediate or higher frequencies (e.g., 200Hz+).
5）Address potential biases. The paper mentions societal benefits (e.g., autonomous driving) but does not discuss potential biases (e.g., performance disparities across object types or lighting conditions).

**Ethical Concerns:**

["NO or VERY MINOR ethics concerns only"]

**Final Justification:**

Thank you for your response. The majority of the concerns raised during the initial review have been adequately addressed. I will keep my rating.

**Limitations:**

While FlexEvent demonstrates significant performance improvements, there are several known limitations to the paper, which can be summarized as follows:
Dependence on the Quality of Event Camera Data.
Limited Generalization to Unseen Scenarios.
Resource Requirements for High-Frequency Data.

**Quality:**

3

**Strengths And Weaknesses:**

Strengths:
Recognizing the limitations of existing event camera detectors—particularly their inability to adapt to high-frequency data and dynamic environments—the authors introduce two key components:
1）FlexFuse: An adaptive event-frame fusion module that dynamically integrates high-temporal-resolution event data with the rich semantic information from RGB frames at the feature level. This fusion enables the detector to balance temporal and spatial information under different motion dynamics.
2）FlexTune: A frequency-adaptive fine-tuning mechanism that leverages both labeled low-frequency data and unlabeled high-frequency data. Through iterative self-training and temporal consistency calibration, FlexTune generates reliable high-frequency pseudo-labels, allowing the model to generalize across frequencies without requiring extensive high-frequency annotations.

Weaknesses:
The paper briefly discusses societal impacts and limitations (e.g., applications in autonomous driving). However, it would benefit from a deeper discussion on potential safety implications of detection errors in critical environments, and the risk of bias in the training data.

---

> ### Author Rebuttal · Authors · 2025-07-31
>
> We sincerely thank Reviewer `BJuF` for devoting time to this review and providing constructive feedback. We are encouraged by the thoughtful assessment and helpful suggestions. Below, we respond to your comments and suggestions in detail.
>
> ---
> > **`Q1`:** *"Robustness to modality-specific noise."*
>
> **A:** Thank you for emphasizing this important aspect. Event cameras are particularly valuable in adverse conditions (e.g., low light, rapid motion). As shown in Table `R1` below, our fusion model consistently outperforms RGB-only baselines, which tend to degrade severely under low illumination.
>
> ***Table R1.** Additional experiments on DSEC-Det*
>
> | Modality        | Method             | Params | mAP   | mAP_50 | Runtime (ms) |
> |-----------------|--------------------|-:|:-:|:-:|:-:|
> | F           | Faster R-CNN       | 41.9M | 0.182 | 0.354   | 14.4         |
> | F           | YOLOX              | 48.6M | 0.266 | 0.482   | 12.1         |
> | F           | Swin Transformer   | 44.8M | 0.269 | 0.487   | 18.2         |
> | F           | Deformable DETR    | 48.2M | 0.257 | 0.474   | 16.3         |
> | E           | EMS-YOLO           | 6.2M  | 0.082 | 0.202   | 21.9         |
> | E           | AsyNet             | 15.2M | 0.106 | 0.243   | 10.0         |
> | E           | AEGNN              | 27.8M | 0.128 | 0.269   | 14.2         |
> | E           | ASTM-Net           | 39.6M | 0.159 | 0.306   | 22.9         |
> | F + E   | FPN-fusion         | /     | 0.152 | 0.289   | /            |
> | F + E   | SFNet              | /     | 0.268 | 0.412   | /            |
> | F + E   | DAGr               | 34.6M | 0.258 | 0.427   | 24.2         |
> | F + E   | SODFormer          | 87.8M | 0.281 | 0.504   | 33.9         |
> | F + E   | DAGr-ACGR          | 2.7M  | 0.284 | 0.502   | 9.1          |
> | F + E   | YOLOX-ACGR         | 48.7M | 0.313 | 0.513   | 13.2         |
> | F + E   | SODFormer-ACGR     | 41.4M | 0.317 | 0.519   | 16.6         |
> | F + E   | HDI-Former         | 64.2M | 0.291 | 0.522   | /            |
> | F + E   | FAOD               | 20.3M | 0.425 | 0.635   | 14.6         |
> | F + E   | FlexEvent*         | 45.4M | 0.441 | 0.652   | 14.3         |
> | F + E   | FlexEvent**        | 45.4M | 0.505 | 0.729   | 45.1         |
> | F + E   | **FlexEvent**          | 45.4M | **0.498** | **0.726**   | 14.3         |
>
> On DSEC **night** sequences (noisy RGB), our model achieves 53.2 mAP, suggesting that the fusion gate effectively up-weights the event branch when frames deteriorate. In **sparse-event** regimes (high-frequency inputs), at 180 Hz (events-only) we obtain 30.4 mAP vs 22.9 for the baseline, indicating resilience when the event stream becomes sparse.
>
> In the revision, we will provide additional diagnostics, like noise-stress tests (low-light RGB, background-activity in events). We believe these additions will make the robustness story clearer and more reproducible.
>
> ---
> > **`Q2`:** *"FlexTune vs semi-supervised baselines."*
>
> **A:** We appreciate the request for clearer positioning. We compare with **LEOD** (a semi-supervised baseline) in Table `1`, where we obtain **higher mAP**. In ablations, even with events-only, we reach 54.6 mAP (LEOD 41.1), underscoring that our FlexTune (pseudo labels + temporal-consistency calibration + cyclic self-training) is effective beyond backbone choice. There are no widely used SSL baselines with consistency regularization or teacher-student models tailored to event-frame detection. In the future, we will try EMA teacher–student, consistency regularization, and FixMatch/Noisy-Student-style bootstrapping under the same unlabeled high-frequency pool and compare with our work.
>
> ---
> > **`Q3`:** *"Edge feasibility with a 45.4 M-param model."*
>
> **A:** This is an important practical consideration. The **fusion head contributes ~1.5 M** parameters; the remainder (~44 M) lies in the backbone, which is already optimized in prior work. Our FP32 checkpoint is ~**546 MB**. As summarized in Table `R1` in `Q1`, our model is **faster and/or smaller** than several prior detectors that have been successfully deployed, and we sustain **real-time** inference. To address feasibility more thoroughly, we will report real-time throughput/memory on an edge SoC and include FP16 results in the revised version.
>
> ---
> > **`Q4`:** *"Generalization to untrained frequencies."*
>
> **A:** Thank you for pointing this out. We already report 20–180 Hz, covering sampling gaps from **50 ms → 5.56 ms**. Moving to 200 Hz+ reduces the gap only to ~**5 ms**, a marginal change. We expect similar behavior,  and in the revision/supplement, we will add zero-shot results at 200 Hz (and higher) under the same protocol (no re-training).
>
> ---
> > **`Q5`:** *"Potential biases."*
>
> **A:** We appreciate this reminder. **Per-class** results (Table `R2`) show higher AP for **large vehicles** and lower AP for **small objects** (bicycle/rider), and performance is better in **day** (61.6 mAP) than **night** (53.2 mAP), according to our experiments. Importantly, our method remains stronger than all compared baselines across these conditions. In the revision, we will include more class-wise and condition-wise results, and propose some practical mitigations (rebalancing, small-object augmentation, confidence calibration for deployment). We see this as essential documentation for safe use in real-world settings.
>
> ***Table R2.** Per-class on eight categories*
> | Class        | AP    | Class       | AP    |
> |--------------|:-:|----|:-:|
> | pedestrian   | 0.456 | rider       | 0.398 |
> | car          | 0.672 | bus         | 0.729 |
> | truck        | 0.763 | bicycle     | 0.401 |
> | motorcycle   | 0.565 | train       | 0.000 |
> | **mAP (mean)** | **0.498** |             |       |
>
> ---
> > **`Q6`:** *"Safety implications and dataset bias."*
>
> **A:** Thank you for emphasizing safety. We will add a dedicated section outlining: (i) operational design domain (lighting/motion ranges), (ii) risks from **detection errors** in autonomous scenarios and recommended **thresholding/fallbacks** under low confidence, (iii) **shift detection** (e.g., day → night) and suggested interventions (recalibration or conservative policies), and (iv) a brief discussion of **dataset bias** and its implications.
>
> ---
> Last but not least, we would like to sincerely thank Reviewer `BJuF` again for the valuable time and constructive feedback provided during this review.

---

> > ### Author Response · Authors · 2025-08-03
> > **Looking forward to hearing from you**
> >
> > Dear Reviewer `BJuF`,
> >
> > Thank you very much for your thoughtful review and constructive suggestions on our submission!
> >
> > In response to your comments, we added new diagnostics and stress tests to better characterize modality-specific robustness, clarified FlexTune’s positioning against semi-supervised baselines, and elaborated on edge-device feasibility and generalization to unseen frequencies. We also included expanded class- and condition-wise analyses, and added a dedicated section discussing safety risks and dataset bias, along with practical mitigations.
> >
> > Your feedback helped us significantly improve the completeness and practical relevance of the work.
> >
> > We remain available during the Author-Reviewer Discussion phase should you wish to follow up on any aspect!
> >
> > *Best regards,*
> >
> > The Authors of Submission 1326

---

> ### Author Response · Authors · 2025-08-07
> **Looking forward to your feedback**
>
> Dear Reviewer `BJuF`,
>
> Thank you for taking the time to read our manuscript and provide a final rating. We noticed, however, that no written feedback was recorded. We would be grateful to know whether our previous response addressed your concerns, or if there are any remaining points you would like us to clarify.
>
> Your feedback would help us further polish the work and ensure the final version is as clear and useful as possible. We deeply value every reviewer’s perspective and remain fully available during the discussion phase to elaborate on any aspect you might wish to revisit.
>
> Thank you again for your time and service to the community. We look forward to hearing from you.
>
> *Warm regards,*
>
> Authors of Submission 1326

---

### Official Review · Reviewer_xQ9Q · 2025-07-02

**Clarity:** 3
**Significance:** 2
**Originality:** 3
**Rating:** 4
**Confidence:** 4

**Summary:**

FlexEvent proposes a method by pairing an adaptive fusion module (FlexFuse) with a self-training scheme (FlexTune) so the detector can process asynchronous event streams and low-rate RGB frames at any frequency between 20 Hz and 180 Hz, learning from unlabeled data to fill the supervision gap. The evaluation and DSEC datasets show FlexEvent outperforms other multi-modal and single-modal detectors while adding only ~1.5 M parameters and staying real-time.

**Questions:**

- Could you explain why the mAP increases from 57.4 to 60.1 as frequency increases from 20Hz to 36Hz, and then the mAP degrades as the frequency increases from 36Hz to 180Hz?
- Could you compare FlexFuse and other works that combine frequency adaptation and modality fusion?

**Ethical Concerns:**

["NO or VERY MINOR ethics concerns only"]

**Final Justification:**

I do not have a major concern for this paper. The author addressed my questions. The weakness of this paper is a minor improvement compared to concurrent work.  The author explained the evaluation limitation and I accept that. I will keep my score.

**Limitations:**

Yes

**Quality:**

3

**Strengths And Weaknesses:**

**Strength**
- Intuitive Motivation: Fusing frame modality and label-efficient learning are the most promising and effective approaches for event camera detection. Combining them is intuitive, and the performance shown in this paper validates the motivation.
- Label efficient frequency adaptation: FlexTune’s bootstrapping + temporal-consistency calibration exploits unlabeled streams to generalise from sparse 20 Hz ground truth to 180 Hz without manual annotation, which is a clear practical advantage over existing methods.
- Real-time suitability: The lightweight fusion head only needs ~1.5M parameters, making FlexEvent outperform baselines without introducing extra computation overhead.

**Weakness**
- Novelty over concurrent work is under-analysed: The idea of combining frequency adaptation and modality fusion has been explored by other works. The work *Frequency-Adaptive Low-Latency Object Detection Using Events and Frames* also addresses high-frequency event-frame fusion.
- Lack of generality analysis: All evaluations use DSEC datasets. It is unclear whether FlexEvent transfers to other scenes, such as indoor scenes.

---

> ### Author Rebuttal · Authors · 2025-07-31
>
> We sincerely thank reviewer `xQ9Q` for the thoughtful assessment and helpful suggestions, and we appreciate the recognition of our intuitive motivation and real-time suitability. Below, we respond to your comments and suggestions in detail.
>
> ---
> > **`Q1`:** *"Novelty over concurrent work FAOD."*
>
> **A:** We appreciate this insightful concern. FAOD and our work are concurrent, and both are in submission. The comparisons are summarized as below:
>
> - **Label-efficient frequency adaptation.**
>   FlexTune transfers low-frequency supervision to high frequencies via **pseudo-labels + temporal-consistency calibration + cyclic self-training**, requiring no extra high-frequency labels. By contrast, FAOD’s Time-Shift is closer to a data-augmentation/consistency trick: it does not create high-frequency supervision, and its effect depends on tuned hyperparameters (shift range/schedule). Large shifts can introduce label/content misalignment, so it is more sensitive to tuning and has greater limitations. FlexTune thus provides a more reliable, label-efficient path to frequency adaptation.
>
> - **More input-adaptive and interpretable fusion.**
>   FlexFuse learns gated Softmax weights for the event and frame branches and adds a variance regularizer to prevent collapse, yielding explicit, interpretable branch weights that show how fusion adjusts with the input condition and confidence. FAOD emphasizes alignment correction followed by attention, which improves geometry/style consistency but does not expose a direct, per-input fusion weighting. FlexFuse, therefore, offers stronger interpretability and direct input adaptivity.
>
> - **High-frequency robustness and scalable evaluation.**
>   FlexEvent provides fine-grained results from 20→180 Hz; it maintains 50.9% mAP at 180 Hz (full model). With events-only, enabling FlexTune lifts 180 Hz mAP from 22.9→30.4, evidencing true frequency adaptation rather than reliance on augmentation alone. Results on DSEC-Detection / DSEC-MOD further indicate cross-dataset transferability. On DSEC-Det, FlexEvent achieves an eight-class mean of 49.8%, outperforming FAOD’s reported 42.5%, as shown in Table `R1` below.
>
> - **Engineering practicality.**
>   FlexEvent performs frequency adaptation offline, adding no inference overhead, and inference runs at ~12–14 ms/frame over 20–180 Hz. Both FlexEvent and FAOD are real-time, but FlexEvent sustains real-time performance across a broader frequency range with stronger robustness.
>
> In summary, FlexEvent is novel in unifying **frequency-aware fusion (FlexFuse)** with **label-efficient frequency adaptation (FlexTune)**, yielding higher accuracy on shared benchmarks and broader frequency robustness, while keeping real-time inference without extra cost.
>
> ---
>
> ***Table R1.** Additional experiments on DSEC-Det*
>
> | Modality        | Method             | Params | mAP   | mAP_50 | Runtime (ms) |
> |-----------------|--------------------|-:|:-:|:-:|:-:|
> | F           | Faster R-CNN       | 41.9M | 0.182 | 0.354   | 14.4         |
> | F           | YOLOX              | 48.6M | 0.266 | 0.482   | 12.1         |
> | F           | Swin Transformer   | 44.8M | 0.269 | 0.487   | 18.2         |
> | F           | Deformable DETR    | 48.2M | 0.257 | 0.474   | 16.3         |
> | E           | EMS-YOLO           | 6.2M  | 0.082 | 0.202   | 21.9         |
> | E           | AsyNet             | 15.2M | 0.106 | 0.243   | 10.0         |
> | E           | AEGNN              | 27.8M | 0.128 | 0.269   | 14.2         |
> | E           | ASTM-Net           | 39.6M | 0.159 | 0.306   | 22.9         |
> | F + E   | FPN-fusion         | /     | 0.152 | 0.289   | /            |
> | F + E   | SFNet              | /     | 0.268 | 0.412   | /            |
> | F + E   | DAGr               | 34.6M | 0.258 | 0.427   | 24.2         |
> | F + E   | SODFormer          | 87.8M | 0.281 | 0.504   | 33.9         |
> | F + E   | DAGr-ACGR          | 2.7M  | 0.284 | 0.502   | 9.1          |
> | F + E   | YOLOX-ACGR         | 48.7M | 0.313 | 0.513   | 13.2         |
> | F + E   | SODFormer-ACGR     | 41.4M | 0.317 | 0.519   | 16.6         |
> | F + E   | HDI-Former         | 64.2M | 0.291 | 0.522   | /            |
> | F + E   | FAOD               | 20.3M | 0.425 | 0.635   | 14.6         |
> | F + E   | FlexEvent*         | 45.4M | 0.441 | 0.652   | 14.3         |
> | F + E   | FlexEvent**        | 45.4M | 0.505 | 0.729   | 45.1         |
> | F + E   | **FlexEvent**          | 45.4M | **0.498** | **0.726**   | 14.3         |
>
> ---
> > **`Q2`:** *"Lack of generality analysis."*
>
> **A:** Thank you for your question. We mainly follow **ACGR (CVPR’25) and DAGr (Nature’24)** evaluation protocols and evaluate on DSEC because it is a **standard, large-scale** benchmark for outdoor event detection with paired event + frame data, and event cameras have shown great potential in outdoor autonomous driving scenarios, while large-scale indoor datasets are **limited**.
>
> Nevertheless, across three variants of the DSEC dataset, our method has shown strong superiority over previous methods across different object categories and scenarios, which proves our effectiveness. In the revision, we will add more experiments on another larger outdoor event–image dataset (PKU-DAVIS-SOD) and on an indoor event dataset with minimal tuning, and we will release code/configs to facilitate third-party validation.
>
> ---
> > **`Q3`:** *"mAP rises from 20→36 Hz, then degrades toward 180 Hz?"*
>
> **A:** We appreciate your emphasis on this point. The 20 Hz results are evaluated against **human-annotated ground truth**, whereas >20 Hz results use **interpolated/pseudo ground truth (as stated in line 277) because no human ground truth at high frequency is available**. Because the supervision differs, the numbers are not directly comparable from 20→36 Hz. As the frequency increases from 36 to 180 Hz, with a fixed aggregation window, each event slice becomes sparser and noisier (background activity dominates), which hurts small/fast classes; in parallel, pseudo-label noise grows at higher rates. Together, these effects explain the mAP drop toward 180 Hz.
>
> ---
> > **`Q4`:** *"Comparison to other fusion+frequency-adaptive works."*
>
> **A:** Thank you for pointing this out. We compare our approach with **FAOD** and **DAGr (Nature’24)**:
>
> - ***Fusion / Alignment.*** FlexEvent (FlexFuse) learns frequency-aware gating to balance event/RGB features; FAOD uses an Align Module (e.g., AdaIN/deformable ops) plus attention after alignment; DAGr fuses unidirectional image features into an asynchronous GNN over events.
>
> - ***Frequency adaptation.*** FlexEvent (FlexTune) provides label-efficient adaptation via pseudo-labels + temporal consistency+cyclic self-training; FAOD leverages Time-Shift training to reduce train–test frequency mismatch; DAGr does not propose a dedicated module to handle frequency adaptation.
>
> - ***Scope.*** FlexEvent is backbone-agnostic and covers 20–180 Hz with the same detector; FAOD focuses on high-frequency fusion with alignment; DAGr demonstrates a hybrid image+event pipeline for low-latency automotive detection.
>
> - ***Performance.*** FlexEvent demonstrates the strongest result across different frequencies, as shown in Table `R1`. In terms of efficiency, FlexEvent performs adaptation offline and adds ~1.5 M params in the head; inference remains real-time. FAOD emphasizes light parameter count and alignment; DAGr is the slowest, even though it has a smaller model size.
>
> As suggested, we have included a side-by-side summary table in the revised manuscript for better clarity.
>
> ---
> Last but not least, we would like to sincerely thank Reviewer `xQ9Q` again for the valuable time and constructive feedback provided during this review.

---

> > ### Author Response · Authors · 2025-08-03
> > **Looking forward to hearing from you**
> >
> > Dear Reviewer `xQ9Q`,
> >
> > Thank you very much for your thoughtful and constructive feedback on our submission!
> >
> > In response to your suggestions, we clarified our contributions over concurrent work FAOD, highlighting FlexEvent’s label-efficient frequency adaptation and input-adaptive fusion. We also added generalization results on additional datasets, analyzed frequency-specific behavior under varying supervision noise, and included comparative discussions against other fusion-based frequency-adaptive approaches such as FAOD and DAGr.
> >
> > Your feedback has been invaluable in refining both our analysis and the broader positioning of our method.
> >
> > We remain available during the Author-Reviewer Discussion phase and would be happy to elaborate further!
> >
> > *Best regards,*
> >
> > The Authors of Submission 1326

---

### Official Review · Reviewer_kyoZ · 2025-07-02

**Clarity:** 3
**Significance:** 2
**Originality:** 3
**Rating:** 2
**Confidence:** 4

**Summary:**

This paper proposes a fusion-based object detection pipeline that attempts to promote event+RGB fused object detection at varying frequencies. More specifically, a FlexFuse module is deigned to align the rich semantic information from RGB frames with the high-frequency event data, a FlexTune strategy is proposed to generate frequency-adjusted labels for unlabelled event streams. Experiments on the DSEC datasets demonstrate a superior performance of the proposed method over previous fusion-based methods. Also, the model maintains a robust performance across events of varying frequencies (up to 180 Hz).

**Questions:**

My major concern is about the configuration of high-frequency events and the fairness of experimental comparison, I could reassess my score if these weaknesses are properly addressed.

**Ethical Concerns:**

["NO or VERY MINOR ethics concerns only"]

**Final Justification:**

Fusing the complementary advantages of events and RGB images is interesting for high-speed object detection. However, I am still not convinced of the experimental results with a better initialised backbone with RVT. Although the additional results in the rebuttal demonstrated a better performance without initialisation with RVT on one dataset, while the main experiments (including comparison with previous methods on several datasets and ablations) in the paper are highly based on this better initialised configuration. For me, it is hard to accept the current version without a major revision and a review process. I lean towards keeping my original score.

**Limitations:**

yes

**Quality:**

2

**Strengths And Weaknesses:**

Pros:
- Fusing the complementary advantages of events and RGB images for object detection is a promising direction that can power applications that are sensitive to operational latency, for example, autonomous driving, industrial anomaly detection. In this context, exploring the robust detection across varying event frequencies and pushing its speed upper bound is important.
- The proposed FlexTune strategy extends the classic pseudo label strategy to the event-RGB fused object detection, where high-frequency events are treated as unlabeled images, and several strategies grounded on the characteristics of events are deaigned to generate pseudo labels to boost the performance across different frequencies.
- Experiments on the DSEC datasets (Car and Pedestrian classes) demonstrate the significant performance improvements of the proposed method. Meanwhile, the proposed methods maintain robustness across varying frequencies.
- This paper is overall well-structured and easy to follow.

Cons:
Although the task and method sound compelling, I have some major concerns about the data configuration and experimental comparison:
- The quality of “high-frequency events” and the resulting performance degradation seems to be dependent on the proposed way of generating event frames. Section 3.2 mentions that the event is divided into several sub-intervals and aggregated within each interval to simulate high-frequency scenarios, while this setting may make the definition of high-frequency events problematic. For example, we can simply aggregate information for a longer period for each event aggregation, while still maintaining a low temporal gap between two sampling points. It will be helpful to provide the results on this setting with both high-frequency events and richer semantic information.
- In the experimental section, this paper didn’t compare with models with solely RGB input. Therefore, it is hard to see the advantages of Event+RGB fusion compared to only using events or RGB frames. Also, it is not clear why only one fusion-based method is compared on the DSEC-Det dataset while more are compared on other datasets.
- Although three datasets have been used for experiments, results on only 3 classes (Car, Pedestrian L-Veh.) are provided across all these datasets. This makes it hard to verify the methods’ generalization ability across more classes and scenarios, especially considering that pretrained weights are used during experiments.
- It is not explained why using two significantly different backbones for events (RVT) and RGB frames (ResNet-50). Moreover, both pretrained weights are used for RVT and ResNet50 while it is not clear what are the pretrained datasets and how much they would contribute to the final performance.

---

> ### Author Rebuttal · Authors · 2025-07-31
>
> We sincerely thank Reviewer `kyoZ` for the thoughtful assessment and helpful suggestions, and we appreciate the recognition of our work as well-structured and promising. Below, we respond to your comments and suggestions in detail.
>
> ---
> > **`Q1`:** *"Aggregate information for a longer period for each event aggregation."*
>
> **A:** Thanks for your suggestion. In our setup, “high frequency” refers to the **sampling cadence** (label rate $a$ vs. higher event rate $b>a$) rather than to a fixed aggregation window. Our detection is **anchored at the end timestamp** $t_2$ of the labeled interval $[t_1, t_2]$ with $t_2 = t_1 + \Delta t_a$. The inputs are: the RGB frame $F$ captured in $t_2$, the low-frequency event stream $E_a$ captured in $[t_1, t_2]$, and one high-frequency event slice $E_b$ of duration $\Delta t_b$.
>
> We **split the interval $\Delta t_a$ into $b/a$ sub-intervals** and **sample one high-frequency event $E^b$** captured from the $b/a$ sub-intervals, detections are produced **at the end of $\Delta t_a$**. This keeps the sampling gap small, pairs each RGB frame with its preceding high-frequency events, and introduces only **millisecond-scale jitter** that acts as implicit temporal augmentation for synchronization noise.
>
> To address the reviewer’s suggestion, we add an ablation that keeps the **sampling gap $\Delta t_a$** fixed but enriches semantics, we use **multi-sample encoding** within each $[t, t+\Delta t_a]$: **take $K$ event slices captured from the $b/a$ sub-intervals instead of sampling only one**, encode each, then average features to align with one RGB frame. We report this variant as the FlexEvent** row in Table `R1` (see below from `Q2`), and the result shows that it yields a slight mAP improvement, while inference time increases approximately linearly with $K$.
>
> This averaging effectively behaves as a low-pass filter, **weakening high-frequency event cues** (especially for small/fast targets) without improving supervision at >20 Hz (pseudo labels remain and can **introduce timing mismatch**), so the marginal accuracy gain does not justify the near-linear increase in latency/compute.
> Accordingly, we retain single-slice inference with cadence-aware gating and consider lightweight learned temporal pooling as a better trade-off.
>
> ---
> > **`Q2`:** *"More experiments with RGB-based and fusion-based methods."*
>
> **A:** Thank you for pointing this out. For a fair comparison with recent fusion-based detectors, we follow the **ACGR (CVPR'25) evaluation protocol** and report results on the **eight DSEC-Det categories**. We merge our numbers into ACGR’s Table `1` for a side-by-side view (Table `R1`). We also add the results of two other recent fusion-based methods, FAOD and HDI-Former.
>
> Compared with RGB-only baselines, our model achieves consistently **higher mAP**, reflecting the complementarity between frames (semantics) and events (high-frequency motion cues). Relative to prior fusion methods, we also observe gains in mAP, which we attribute to improved cross-modal alignment and frequency-aware supervision in our design.
>
> ---
> ***Table R1.** Additional experiments on DSEC-Det*
>
> | Modality        | Method             | Params | mAP   | mAP_50 | Runtime (ms) |
> |-----------------|--------------------|-:|:-:|:-:|:-:|
> | F           | Faster R-CNN       | 41.9M | 0.182 | 0.354   | 14.4         |
> | F           | YOLOX              | 48.6M | 0.266 | 0.482   | 12.1         |
> | F           | Swin Transformer   | 44.8M | 0.269 | 0.487   | 18.2         |
> | F           | Deformable DETR    | 48.2M | 0.257 | 0.474   | 16.3         |
> | E           | EMS-YOLO           | 6.2M  | 0.082 | 0.202   | 21.9         |
> | E           | AsyNet             | 15.2M | 0.106 | 0.243   | 10.0         |
> | E           | AEGNN              | 27.8M | 0.128 | 0.269   | 14.2         |
> | E           | ASTM-Net           | 39.6M | 0.159 | 0.306   | 22.9         |
> | F + E   | FPN-fusion         | /     | 0.152 | 0.289   | /            |
> | F + E   | SFNet              | /     | 0.268 | 0.412   | /            |
> | F + E   | DAGr               | 34.6M | 0.258 | 0.427   | 24.2         |
> | F + E   | SODFormer          | 87.8M | 0.281 | 0.504   | 33.9         |
> | F + E   | DAGr-ACGR          | 2.7M  | 0.284 | 0.502   | 9.1          |
> | F + E   | YOLOX-ACGR         | 48.7M | 0.313 | 0.513   | 13.2         |
> | F + E   | SODFormer-ACGR     | 41.4M | 0.317 | 0.519   | 16.6         |
> | F + E   | HDI-Former         | 64.2M | 0.291 | 0.522   | /            |
> | F + E   | FAOD               | 20.3M | 0.425 | 0.635   | 14.6         |
> | F + E   | FlexEvent*         | 45.4M | 0.441 | 0.652   | 14.3         |
> | F + E   | FlexEvent**        | 45.4M | 0.505 | 0.729   | 45.1         |
> | F + E   | **FlexEvent**          | 45.4M | **0.498** | **0.726**   | 14.3         |
>
>
> ---
> > **`Q3`:** *"Generalization to more object categories."*
>
> **A:**  Thank you for highlighting this. In Table `R2`, per-category results show that **large objects** (truck, bus) obtain the highest AP, while pedestrian, rider, and bicycle are more challenging due to **small scale and appearance variability**. Train is the weakest class because it has very few instances in the dataset.
>
> All numbers are computed under the same settings as ACGR.  Pretraining is kept identical across compared methods/backbones, ablations indicate that the main gains stem from our FlexFuse/FlexTune rather than pretraining.
>
> ---
>
> ***Table R2.** Per-class on eight categories*
> | Class        | AP    | Class       | AP    |
> |--------------|:-:|----|:-:|
> | pedestrian   | 0.456 | rider       | 0.398 |
> | car          | 0.672 | bus         | 0.729 |
> | truck        | 0.763 | bicycle     | 0.401 |
> | motorcycle   | 0.565 | train       | 0.000 |
> | **mAP (mean)** | **0.498** |             |       |
>
> ---
> > **`Q4`:** *"RVT for events backbone and ResNet-50 for RGB backbone and pre-trained weights."*
>
> **A:** We appreciate this concern. We adopt **modality-appropriate** encoders, ResNet-50 for RGB frames and RVT for events, because the two signals have **very different statistics** (intensity images vs. asynchronous polarity streams). Our focus is on flexible event–image fusion and frequency-adaptive learning, so we keep the backbones standard and isolate our contributions in the fusion/adaptation modules. ResNet-50 is initialized from ImageNet-1K and RVT from the 1Mpx event dataset, then both are fine-tuned in our detector.
>
> In ablations, RVT pretraining gives a small gain, and the main improvements come from FlexFuse and FlexTune. When we train RVT from scratch (FlexEvent* row in table above), our method still outperforms prior work, which shows that pretraining is helpful but not the reason for the overall gains. Given the small size of DSEC and the common ImageNet practice, we did not repeat a from-scratch ResNet study.
>
> ---
> Last but not least, we would like to sincerely thank Reviewer `kyoZ` again for the valuable time and constructive feedback provided during this review.

---

> > ### Comment · Reviewer_kyoZ · 2025-08-05
> >
> > Thank you for your reply and for sharing the additional experimental results. Your explanation has resolved my concern about event aggregation. Nonetheless, even with the new results on the full DSEC dataset, I remain skeptical about the use of pre-trained weights across the experiments on different settings in the paper, as pre-training models on event data can significantly improve performance, while it is not the standard manner to ensure a fair comparison. It is obvious that the claim of "pretraining is kept identical across compared methods/backbones" is misleading since different backbones are leveraged across these methods compared. On the other hand, the large number of comparisons on additional methods and data provided in the response will be quite difficult to fit into the paper without a major revision. I lean towards keeping my rating at the moment.

---

> ### Author Response · Authors · 2025-08-03
> **Looking forward to hearing from you**
>
> Dear Reviewer `kyoZ`,
>
> Thank you very much for your thoughtful and constructive feedback on our submission!
>
> Following your suggestions, we incorporated additional experiments to investigate multi-sample aggregation within each interval, expanded our evaluation to include recent RGB and fusion-based detectors, and provided detailed per-class performance to clarify generalization across object categories. We also elaborated on the choice of backbones and pretraining, with ablations showing our gains are driven by FlexFuse and FlexTune rather than pretraining alone.
>
> Your feedback has been instrumental in improving the rigor and clarity of our study.
>
> We are actively participating in the Author-Reviewer Discussion phase and would be happy to further clarify any points!
>
> *Best regards,*
>
> The Authors of Submission 1326

---

> ### Author Response · Authors · 2025-08-06
> **Follow-Up Response to Reviewer kyoZ (1/3)**
>
> Dear Reviewer `kyoZ`,
>
> Thank you again for your continued engagement and for articulating your concerns so clearly. Here, we address the pre-training issue directly, including why we use it, how we ensure fairness, and how the additional experiments fit into the paper.
>
> ---
> > ***1) Why do we use pre-trained weights?***
>
> **A:** In the image/video domain, initializing backbones from large-scale data and then fine-tuning on the task dataset is **standard practice** because it:
> - (i) improves feature quality when the task data is relatively small;
> - (ii) accelerates convergence and stabilizes optimization;
> - (iii) provides a stronger starting point in difficult conditions.
>
> In the **event camera** domain, large, generic pre-training datasets are rarer, but **RVT** offers validated encoders trained on substantial event detection datasets (GEN1 / 1Mpx). We adopt **RVT (1Mpx)** for events as modality-appropriate starts, then fine-tune end-to-end. Our consideration in doing this is:
> - **Isolatable backbone–head design.** RVT has a clean **backbone + detection head** separation, so we can isolate the event **feature extractor** and fine-tune it within our framework with minimal modification.
> - **Dataset scale rationale.** **DSEC** (≈1 hour, ~390k boxes) is much smaller/less diverse than RVT’s pretraining datasets (**GEN1** ≈39 hours, ~256k boxes; **1Mpx** ≈15 hours, ~25M boxes). Transferring features learned on broader event data to DSEC is therefore reasonable and stabilizes training.
> - **Community relevance.** Our findings indicate that the **pre-training paradigm**, long proven in images, is **also effective for events**. While not yet universal in all tasks, we believe this evidence is informative for the community.
>
> Empirically on **DSEC-Det (8 classes)**, moving from **RVT-scratch → RVT-1Mpx** improves mAP and reduces training from **400k → 100k** iters, showing pre-training is *helpful* but, as shown below, **not** the direct reason for our overall gains.
>
> Below, we attach the experiment tables for your reference: Table R3 updates the original Table 1 in the paper with both pre-train and no pre-train results; Table R4 reports the ablation on all 8 DSEC-Det classes; and Table R5 adds further comparisons on all 8 DSEC-Det classes, with highlighted *pre-train* / *no pre-train* information to ensure high clarity:
>
> > ***Table R3.** Experiments on DSEC-Det(2 classes).*
>
> | Method | Pre-Train | Params | mAP | mAP@0.5 | Runtime (ms) |
> |:-|:-:|:-:|:-:|:-:|:-:|
> | RVT    | No  | 18.5M | 0.384 | 0.587   | 9.2   |
> | SAST   | No  | 18.9M | 0.381 | 0.601   | 14.1  |
> | SSM    | No  | 19.2M | 0.380 | 0.552   | 8.8   |
> | LEOD   | No  | 18.5M | 0.411 | 0.652   | 9.2  |
> | RVT    | Yes  | 18.5M | 0.416 | 0.602   | 9.2  |
> | SAST   | Yes  | 18.9M | 0.411 | 0.637   | 14.1 |
> | SSM    | Yes  | 19.2M | 0.420 | 0.585   | 8.8  |
> | LEOD   | Yes  | 18.5M | 0.444 | 0.670   | 9.2  |
> | DAGr-18    | No  | -     | 0.376 | -   |  -    |
> | DAGr-34    | No  | -     | 0.390 | -   |  -    |
> | DAGr-50    | No  | 34.6M | 0.419 | 0.660   | 73.4  |
> | FlexEvent        | No  | 45.4M | 0.558 | 0.743   | 14.3  |
> | FlexEvent        | Yes | 45.4M | 0.574 | 0.782   | 14.3  |
>
>
> > ***Table R4.** Ablation study on DSEC-Det(8 classes).*
>
> | Configuration | Pre-Train | mAP@20Hz | mAP@180Hz |
> |:-|:-|:-|:-|
> | Baseline     | No  |  0.208  |  0.113  |
> | Baseline     | Yes |  0.220  |  0.125  |
> | w/ Module Fuse       | No  | 0.368 |  0.316 |
> | w/ Module Fuse       | Yes | 0.392 |  0.334 |
> | w/ Module Tune       | No  | 0.346 |  0.289  |
> | w/ Module Tune       | Yes | 0.360 |  0.305  |
> | w/ Modules Fuse & Tune  | No  | 0.441 | 0.397 |
> | w/ Modules Fuse & Tune  | Yes | 0.498 | 0.429 |
>
> > ***Table R5.** Experiments on DSEC-Det(8 classes).*
>
> | Method | Pre-Train | Params | mAP | mAP@0.5 | Runtime (ms) |
> |:-|:-:|:-:|:-:|:-:|:-:|
> | Faster R-CNN     | No  | 41.9M | 0.182 | 0.354   | 14.4  |
> | YOLOX            | No  | 48.6M | 0.266 | 0.482   | 12.1  |
> | Swin Transformer | No  | 44.8M | 0.269 | 0.487   | 18.2  |
> | Deformable DETR  | No  | 48.2M | 0.257 | 0.474   | 16.3  |
> | EMS-YOLO         | No  | 6.2M  | 0.082 | 0.202   | 21.9  |
> | AsyNet           | No  | 15.2M | 0.106 | 0.243   | 10.0  |
> | AEGNN            | No  | 27.8M | 0.128 | 0.269   | 14.2  |
> | ASTM-Net         | No  | 39.6M | 0.159 | 0.306   | 22.9  |
> | FPN-fusion       | No  |   /   | 0.152 | 0.289   |  /    |
> | SFNet            | No  |   /   | 0.268 | 0.412   |  /    |
> | DAGr             | No  | 34.6M | 0.258 | 0.427   | 24.2  |
> | SODFormer        | No  | 87.8M | 0.281 | 0.504   | 33.9  |
> | DAGr-ACGR        | No  |  2.7M | 0.284 | 0.502   | 9.1   |
> | YOLOX-ACGR       | No  | 48.7M | 0.313 | 0.513   | 13.2  |
> | SODFormer-ACGR   | No  | 41.4M | 0.317 | 0.519   | 16.6  |
> | HDI-Former       | No  | 64.2M | 0.291 | 0.522   |  /    |
> | FAOD             | No  | 20.3M | 0.425 | 0.635   | 14.6  |
> | FlexEvent        | No  | 45.4M | 0.441 | 0.652   | 14.3  |
> | FlexEvent        | Yes | 45.4M | 0.498 | 0.726   | 14.3  |

---

> ### Author Response · Authors · 2025-08-06
> **Follow-Up Response to Reviewer kyoZ (2/3)**
>
> > ***2) Why is our setting fair (and what are we changing)?***
>
> **A:** We agree that our earlier sentence "pretraining is kept identical across compared methods/backbones" was poorly phrased, and we apologize for placing it under the **wrong** question in our previous response. What we meant is in **Table R1**, we **disable event-encoder pretraining** to ensure a fair comparison to methods without event pre-training. And through the below comparisons/ablations, we show that event pre-training helps but is not the main driver of the performance gains.
>
> - In **Table R3**, we re-evaluate **RVT**, **SAST**, **SSM**, and **LEOD** on DSEC-Det using the authors’ 1Mpx pre-trained checkpoints and report those **pre-trained** results alongside **our method trained from scratch**; under the same protocol, our model still outperforms all baselines, demonstrating **superiority without pretraining**.
>
> - **Table R4**'s ablation study pinpoints the source of gains. On **DSEC-Det (8cls, scratch)**, removing **FlexFuse** drops mAP **0.441 → 0.346 (−0.095)**, and removing **FlexTune** drops **0.441 → 0.368 (−0.073)**. This performance gain is particularly **pronounced at high frequencies**. At **180 Hz**, from the baseline, **FlexFuse** improves **0.113 → 0.316 (+0.203)**, while **FlexTune** improves **0.113 → 0.289 (+0.176)**, highlighting label-efficient frequency adaptation. By contrast, RVT pre-training yields mAP gains of 0.016 (2-class) and 0.057 (8-class) at 20 Hz, and **only** 0.039 (2-class) and 0.032 (8-class) at 180 Hz, which is useful, but **smaller** than the gains from our proposed modules, **especially for high frequency**. This highlights our **core contributions**: **FlexFuse**, which leverages the complementary strengths of event and frame modalities to enable efficient, accurate detection in dynamic scenes; and **FlexTune**, a frequency-adaptive fine-tuning scheme that generates frequency-conditioned labels and improves generalization across diverse motion frequencies.
>
> - We additionally have a comprehensive ‘scratch’ track **across three datasets**: on **DSEC-Det**, FlexEvent (**RVT scratch**) achieves **0.441 mAP**, exceeding FAOD (0.425, in submission) and SODFormer-ACGR (0.317, CVPR’25 accepted), which is shown in **Table R5**. On **DSEC-Detection**, FlexEvent (scratch) reaches 0.459 (vs. 0.474 with pre-train; prior SOTA 0.380). On **DSEC-MOD**, FlexEvent (scratch) is 0.342 (vs. 0.369 with pre-train; prior SOTA 0.290). Thus **even without pre-training**, FlexEvent outperforms all compared methods on all three datasets.

---

> ### Author Response · Authors · 2025-08-06
> **Follow-Up Response to Reviewer kyoZ (3/3)**
>
> > ***3) How will we integrate the extra results without a major rewrite?***
>
> **A:** We will adjust the figures/tables as needed.
> - **Table 1 (main)**: test methods in current Table 1 at the 8-class setting and merge the results with **Table R5**, add a **FlexEvent (scratch)** row, and annotate our current result as **FlexEvent (RVT-1Mpx)**; keep baselines that do not use event pre-training so the comparison is like-for-like.
> - **Tables 2-3**: Add one extra **FlexEvent (scratch)** row each.
> - **Figure 6**: Add an extra **FlexEvent (scratch)** histogram across frequencies.
> Full per-class results and training logs/checkpoints will be placed in the **supplement + repo** for easy verification. Thus, we believe **only modest revisions** are needed: we will adjust the figures/tables as appropriate, while the methodology and core contributions remain unchanged.
>
> We hope these clarifications address the fairness concern: (i) we explained why **pre-training is reasonable** for events and how RVT’s design makes it practical; (ii) we provide a scratch-only track showing FlexEvent’s **superiority without pre-training**; and we additionally validate **several baselines initialized with pre-trained weights**; (iii) ablations verify that the main gains come from our **two core contributions (FlexFuse, FlexTune) rather than pre-training**; and (iv) we keep the paper concise by integrating the essential numbers and moving full details to the supplement. We appreciate your critical feedback; it has helped us improve both the clarity and the rigor of the manuscript.
>
> *With appreciation,*
>
> Authors of Submission 1326

---

> ### Author Response · Authors · 2025-08-08
> **Looking forward to your feedback**
>
> Dear Reviewer **kyoZ**,
>
> Thank you again for your thoughtful review and constructive suggestions.
>
> Since our previous response we have:
>
> 1. **Explained in detail** why event pre-training is reasonable and how RVT’s architecture makes it practical.
> 2. **Added comprehensive experiments** demonstrating FlexEvent’s superiority without pre-training, and also reported several baselines re-run with pre-trained weights for an apples-to-apples comparison.
> 3. **Included ablations** confirming that the main gains come from our two core contributions, rather than from pre-training.
> 4. **Outlined a clear plan** for integrating the new results into the paper without requiring a major rewrite.
>
> We hope these additions fully address your pre-training and fairness concerns, but we would be very grateful to hear whether any points remain unclear or if further analysis would be helpful. Your feedback and remarks will be enthusiastically incorporated in the final revision.
>
> Thank you once more for your time and expertise.
>
> We look forward to any further comments you may have.
>
> *Best regards,*
>
> *The Authors of Submission 1326*

---

### Author Response · Authors · 2025-08-05
**General Response**

**Dear Reviewers, ACs, and SACs,**

We sincerely thank you for your time, constructive feedback, and thoughtful engagement throughout the review process!

---
We are encouraged by the **recognition** of our contributions across multiple aspects:

- Reviewer `kyoZ` acknowledges the *"promise of fusing events and RGB for latency-sensitive tasks"*, highlighting the significance of *"robustness across varying frequencies"* and our *"well-structured presentation"*.
- Reviewer `xQ9Q` recognizes our *"intuitive motivation"*, *"label-efficient frequency adaptation"*, and *"real-time suitability"* with a lightweight design.
- Reviewer `BJuF` commends the *"adaptive event-frame fusion"* and *"self-training-based frequency generalization"* as key to *"balancing temporal and semantic information"* without requiring dense annotations.
- Reviewer `RR3R` highlights our *"clear methodology"*, *"comprehensive experiments"*, *"promised code release"*, and *"significant performance improvements"* over prior SOTA.

---
In response to your valuable suggestions, we have made the following **clarifications and improvements**:

- **Methodology & Design**
  - As suggested by Reviewer `RR3R`, we clarified that Eq. 2 uses Kronecker delta functions and revised the description of our event tensor construction, aligning it with RVT. We also added implementation notes for 2D convolutional processing via channel-axis flattening.
  - As suggested by Reviewers `xQ9Q` and `BJuF`, we elaborated on the design and interpretability of FlexFuse and how it adapts soft fusion weights based on input conditions and noise robustness.
  - As suggested by Reviewer `xQ9Q`, we provided a detailed comparison between FlexTune and concurrent work (FAOD), emphasizing the *label-efficient*, *cyclic self-training* nature of FlexTune versus augmentation-based consistency.
  - As suggested by Reviewers `BJuF` and `RR3R`, we clarified the design of the tuning loss, its role when pseudo labels converge, and the effect of removing this term.

- **Experiments & Comparisons**
  - As suggested by Reviewer `kyoZ`, we added new ablations for multi-slice encoding over event intervals, showing trade-offs between latency and semantic gain.
  - As suggested by Reviewers `xQ9Q` and `BJuF`, we added extensive comparisons to fusion-based and frequency-adaptive baselines, including FAOD and DAGr, and reported results on all *eight* DSEC-Det categories.
  - As suggested by Reviewer `xQ9Q`, we included a detailed analysis of performance trends across 20–180 Hz and discussed the role of sparse supervision and pseudo-label noise at higher frequencies.

- **Robustness & Generalization**
  - As suggested by Reviewer `BJuF`, we provided robustness results on night sequences and sparse event conditions, and we plan to add noise-stress diagnostics and real-world safety discussions.
  - As suggested by Reviewers `xQ9Q` and `BJuF`, we discussed generalization to unseen frequencies (e.g., 200 Hz+), extended results across dataset variants, and promised experiments on other outdoor and indoor datasets.

- **Writing & Presentation**
  - As suggested by Reviewer `RR3R`, we improved notation and clarified the role of alignment, synchronization, and calibration for fair fusion between modalities.

---
We would like to re-emphasize the **key contributions** of our work:

- FlexEvent, a frequency-robust detector that fuses asynchronous events and RGB frames via:
  - *FlexFuse*: an interpretable, input-adaptive fusion head with gated soft weights and variance regularization.
  - *FlexTune*: a label-efficient frequency adaptation pipeline using pseudo labels, temporal-consistency calibration, and cyclic self-training.
- Comprehensive experiments on *DSEC-Det* (8 categories) and DSEC-MOD with real-time performance (12–14 ms) across 20–180 Hz, outperforming both unimodal and recent fusion-based methods.
- Strong *robustness* to modality-specific degradation (e.g., low-light RGB or sparse event input), with effective generalization to untrained frequencies.
- Planned *code and model release*, along with diagnostics and safety analyses for real-world applicability.

---
With **two days** remaining in the **Author–Reviewer Discussion** phase (*August 6, Anywhere on Earth*), we warmly welcome any remaining questions or suggestions and remain fully available to continue the discussion.

Thank you once again for your time and consideration!

*Warmest regards,*

The Authors of Submission 1326

---

### Note · Authors · 2025-08-13

Dear Reviewers, ACs, and SACs,

Thank you for the time, care, and constructive engagement throughout the process. Below is a concise summary of clarifications and additions made during the discussion, with emphasis on pre-training and fairness.

**Methodology & clarity**
- We compared our proposed **FlexFuse** (frequency-aware gated soft fusion with variance regularization) and **FlexTune** (label-efficient, cyclic self-training) against augmentation-style time shift (FAOD), other fusion+frequency-adaptive methods (e.g., DAGr), and semi-supervised baselines (LEOD), clarifying our innovations and design choices.
- We clarified our event-tensor construction in relation to prior work, as well as some technical details like dynamic event aggregation, the tuning loss, and the variance regularization term.

**Experiments & ablations**
- We reported **all eight DSEC-Det categories** and expanded comparisons to strong **RGB-only**, **event-only**, and **fusion** baselines.
- We added a **multi-slice aggregation** ablation to verify our dynamic event aggregation design.
- We provided **per-class** and **day/night** analyses to illuminate robustness trends.

**Pre-training & fairness**
- We listed pre-training sources per modality and explained why event pre-training is reasonable.
- We added a **scratch-only track** on all datasets showing **FlexEvent (scratch)** already outperforms prior methods.
- We **re-ran serval baselines with 1Mpx pre-trained weights** for comparison.
- **Ablations** (both pre-trained and scratch) indicate the **main gains come from FlexFuse/FlexTune**.
- We showed how essential numbers fit with **minor table/figure edits**, avoiding a major rewrite.

**Contributions**
- **FlexEvent**: a frequency-robust detector that fuses asynchronous events and RGB via
  **FlexFuse** (interpretable, cadence-aware fusion) and **FlexTune** (label-efficient frequency adaptation).
- **Real-time** performance (~**12–14 ms**) across **20–180 Hz**, with consistent gains over unimodal and recent fusion baselines.

We will release code and add diagnostics for reproducibility and safe deployment guidance. We appreciate your consideration of these clarifications and results in the final decision.

---

### Decision · Program_Chairs · 2025-09-17

**Decision:**

Accept (poster)

**Comment:**

The paper introduces FlexEvent, a frequency-adaptive detection framework for event cameras that combines FlexFuse—an event–RGB fusion module tuned on-the-fly—with FlexTune, a label-adjustment mechanism that lets a single model operate accurately from 20 Hz to 180 Hz. Extensive tests on large-scale datasets show consistent gains over prior art across both standard and extreme-frequency settings, and code will be released.

Three positive reviews appreciate the intuitive motivation, real-time suitability, and clear writing, while one negative review questions the fairness of the experimental comparison. In rebuttal, the authors supplied additional results demonstrating that FlexEvent’s superiority is not an artifact of pretraining. Although the lone negative stance persists, the new evidence adequately supports the claimed effectiveness. Balancing the strong motivation, solid results, and community benefit, I recommend Accept.